



# How to identify design optimization problems that can be improved with a control co-design approach?

Jenna Iori[1], Carlo Luigi Bottasso[2], and Michael Kenneth McWilliam[1]

[1]Department of Wind and Energy Systems, Technical University of Denmark, Frederiksborgvej 399, 4000 Roskilde, Denmark
[2]Wind Energy Institute, Technical University of Munich, 85748 Garching b. München, Germany

**Correspondence:** Jenna Iori (jenio@dtu.dk)

**Abstract.**

Control co-design is a promising approach for wind turbine design due to the importance of the controller in power production, stability and load alleviation. However, the high computational effort required to solve optimization problems with added control design variables is a major obstacle to quantify the benefit of this approach. In this work, we propose a methodology
to identify if a design problem can benefit from control co-design. The estimation method, based on post-optimum sensitivity analysis, quantifies how the optimal objective value varies with a change in control tuning.

The performance of the method is evaluated on a tower design optimization problem, where fatigue load constraints are a major driver, and using a Linear Quadratic Regulator targeting fatigue load alleviation. We use the gradient-based multidisciplinary optimization framework Cp-max. Fatigue damage is evaluated with time-domain simulations corresponding to the
certification standards. The estimation method applied to the optimal tower mass and optimal levelized cost of energy show good agreement with the results of the control-co design optimization, while using only a fraction of the computational effort.

Our results additionally show that there may be little benefit to use control co-design in the presence of an active frequency constraint. However, for a soft-soft tower configuration where the resonance can be avoided with active control, using control co-design results in a higher tower with reduced mass.

**Keywords:** Control co-design, Multi-disciplinary optimization and design, Wind energy, Fatigue alleviation, Wind turbine tower design, LQR control, Design sensitivity analysis

## 1 Introduction

Control co-design (CCD) is a sub-field of dynamic systems design where the controller is designed simultaneously with the structure. Wind turbine design is a promising field of study within CCD because these structures are driven by load constraints,
while at the same time control is important for optimal production and for reducing loads (Garcia-Sanz, 2019; Veers et al., 2022).

Though CCD is not yet widely used in the field of wind energy, several research groups have shown the potential of the method. Chen et al. (2017) include an automatic controller synthesis for the design of a wind turbine blade with individual pitch control and trailing edge flaps, leading to decrease in the levelized cost of energy (LCOE). Deshmukh and Allison





(2016) achieve an 8% improvement in Annual Energy Production (AEP) with a CCD approach compared to a sequential approach, considering torque control only and using a simple set of structural constraints and a linearized model for the turbine dynamics. Pao et al. (2021) report how including control tuning in the design process leads to a cost-effective extreme-scale 13MW downwind turbine rotor. This result was achieved with an iterative design process instead of a fully-coupled approach.

Most wind turbine optimization frameworks rely heavily on steady state analysis (e.g. Zahle et al. (2016)) or a nested/decou-
pled frozen loads approach (e.g. Bottasso et al. (2016)) to reduce the computation effort of the optimization. Yet, CCD requires expensive time domain simulations to be executed within the optimization loop, to assess the effect of changing the control. Such changes to an optimization framework are expensive, both in the code development phase and to execute once completed. This high computational cost makes it difficult to identify designs relevant for CCD, since the design process often requires a trial and error approach. Therefore, a tool is needed to estimate which problems can benefit from CCD without an excessive
computational burden.

From a mathematical point of view, the difference between a CCD and a standard physical design optimization problem can be seen as the addition of the design variables describing the controller action. A promising problem for CCD applications is likely to be sensitive to control tuning. Therefore, we propose a method to estimate how the optimal objective value of a given problem changes when the control changes, in the context of gradient-based optimization. The estimator is built using post-
optimum sensitivity analysis (POSA) (Castillo et al., 2008) on a standard structural optimization problem with fixed control, and can be used to estimate the results of the more complicated CCD optimization. While POSA is not widely used in the field of wind energy, a recent study by McWilliam et al. (2022) uses this approach to identify the design drivers for swept blades.

The proposed estimation method is applied to the design of a wind turbine tower driven by fatigue damage constraints. Several authors have developed control strategies to reduce fatigue damage (Johnson et al., 2012; Camblong et al., 2012),
reducing tower side-side loads by 8% (Kim et al., 2020) and fore-aft fatigue loads by 14% (Nam et al., 2013). Since fatigue damage can be a driving constraints for wind turbine tower (Canet et al., 2021; Dykes et al., 2018), CCD has the potential to improve the design of this component. In the context of CCD however, fatigue reduction is more challenging due to the many long running time-domain simulations that are needed for accurate fatigue calculations. Therefore, an estimation method is particularly relevant for this type of problems before applying CCD directly.

Another important constraint in the design of wind turbine towers is the frequency constraint that prevents resonance with the rotor rotational frequency. Recent development in control design has allowed to design towers without this constraint, called soft-soft towers, where the resonance avoidance is managed by active control. The soft-soft towers generally have a lower mass than standard ones (also called soft-stiff configuration), and their designs can also be driven by fatigue damage (Dykes et al., 2018). In this work, both the standard and soft-soft configurations are studied in order to assess the performance of the
presented estimation method on two different design problems with different sets of constraints.

The paper is organized as follow. Section 2 describes two estimation methods: a first-order estimator taking into account a linear dependency of the problem with control tuning, and a high-order estimator including non-linear effects but subject to additional assumptions. Section 3 describes the tower design problem and control architecture in details, and how to apply the estimator formula in practice. Section 4 compares the estimator to the solution of the corresponding control co-design



optimization problem. Finally, the limitations of this study and potential applications are discussed in Section 5. A nomenclature is provided in Appendix A.

## 2 Methodology

We consider the control co-design Problem 1 below, where $c$ and $x$ represents the control and structural design variables, respectively:

$$\underset{x,c}{\text{minimize}} \quad f(x,c)$$

$$\text{subject to} \quad g_i(x,c) \leq 0 \quad i = 1,...n. \tag{1}$$

In the general case, the objective function $f$ and the constraints $g_i, i = 1,...,n$ depend on both $x$ and $c$. Most existing wind turbine optimization frameworks do not allow to solve Problem 1 directly. Many frameworks are implemented in such a way that the controller design is fixed during the design process. In this context, adding the control design variable $c$ to the existing optimization requires significant development effort. In addition, having the control design variable in the optimization

problem requires to update the time-dependent loads on the structure at each iteration. As a consequence, the computational effort required for the optimization becomes large, and it is generally impractical to attempt to solve the problem.

Instead, it is possible to solve an optimization problem with frozen control, represented by Problem 2, where the control variable is fixed to its reference value $c_r$:

$$\underset{x}{\text{minimize}} \quad z = f(x, c_r)$$

$$\text{subject to} \quad g_i(x, c_r) \leq 0 \quad i = 1,...n. \tag{2}$$

The aim of this work is to understand if the design problem benefits by a CCD approach. In other words, is there sufficient potential improvements to justify the additional effort to solve Problem 1? If Problem 2 can benefit from a CCD formulation, the optimal objective value is likely to be sensitive to a change in the control parameter $c_r$. This means that solving the problem at $c_r$ or $c_r + dc$ will give a significant change in the optimal objective value $dz^*(dc) = z^*(c_r + dc) - z^*(c_r)$. We use post-optimum design sensitivity (Castillo et al., 2008) to estimate $dz^*(dc)$ from the solution of Problem 2.

The change of optimal objective value due to a change of the control parameter $dc$ can be written as a first-order approximation using the gradients of $f$:

$$dz^*(dc) = f(x^* + dx^*, c_r + dc) - f(x^*, c_r) \simeq \nabla_x f(x^*, c_r)^T dx^* + \nabla_c f(x^*, c_r)^T dc. \tag{3}$$

In this equation, the change of optimal solution $dx^*$ is not directly known, but can be characterized with the first-order optimality conditions: the constraints are satisfied and the stationarity condition holds.

First, satisfaction of the constraints means that $g_i(x^* + dx^*, c_r + dc) = g_i(x^*, c_r) = 0$, $i \in \mathcal{I}$, where $\mathcal{I}$ is the set of active constraints. We assume that the active set does not change with $dc$. This equation can be expanded by using a first-order



approximation around point $(\boldsymbol{x}^*, \boldsymbol{c}_r)$ on the left-hand term, resulting in:

$$\nabla_x g_i(\boldsymbol{x}^*, \boldsymbol{c}_r)^T \mathrm{d}\boldsymbol{x}^* = -\nabla_c g_i(\boldsymbol{x}^*, \boldsymbol{c}_r)^T \mathrm{d}\boldsymbol{c}, \quad i \in \mathcal{I}. \tag{4}$$

Then, we can relate the gradient of the constraints to the gradient of the objective function $\nabla_x f(\boldsymbol{x}^*, \boldsymbol{c}_r)$ in Eq. (3) using
the stationarity conditions. For unconstrained optimization, the optimum is a stationarity point of the objective function. This
condition gives practical methods to find the optimum, e.g. with root finding algorithms. However, for constrained optimization,
$\nabla_x f(\boldsymbol{x}^*, \boldsymbol{c}_r) \neq 0$ in general, in the presence of active constraints. In this case, we can characterize the optimum by considering
stationarity points of the Lagrangian function $\mathcal{L}$ instead, also called augmented cost function:

$$\mathcal{L}(\boldsymbol{x}, \boldsymbol{c}_r, \boldsymbol{\lambda}) = f(\boldsymbol{x}, \boldsymbol{c}_r) + \boldsymbol{\lambda}^T \boldsymbol{g}(\boldsymbol{x}, \boldsymbol{c}_r), \tag{5}$$

where $\boldsymbol{\lambda}$ are the Lagrange multipliers. Here, we simplify the problem by considering only the active constraints. For values
of $\boldsymbol{x}$ satisfying the constraints, the value of the Lagrangian function matches the value of the objective function, $\mathcal{L}(\boldsymbol{x}, \boldsymbol{c}_r, \boldsymbol{\lambda}) = f(\boldsymbol{x}, \boldsymbol{c}_r, \boldsymbol{\lambda})$. Then, it is possible to find a set of Lagrange multipliers (noted $\boldsymbol{\lambda}^*$) so that the optimum $\boldsymbol{x}^*$ corresponds to a
stationarity point of $\mathcal{L}$, i.e. $\nabla_x \mathcal{L}(\boldsymbol{x}^*, \boldsymbol{c}_r, \boldsymbol{\lambda}^*) = 0$. Hence, the stationarity condition is obtained:

$$\nabla_x f(\boldsymbol{x}^*, \boldsymbol{c}_r) + \sum_{i \in \mathcal{I}} \lambda_i^* \nabla_x g_i(\boldsymbol{x}^*, \boldsymbol{c}_r) = \boldsymbol{0}. \tag{6}$$

The Lagrange multiplier can be interpreted as the rate of change of the objective function relative to a change in the con-
straint function. For a formal proof of the stationarity condition, the reader is referred to the Karush-Kuhn-Tucker optimality
conditions and textbooks on convex and non-linear optimization (Boyd and Vandenberghe, 2004). Note that the stationarity
condition comes with assumptions on differentiability and strong duality.

The stationarity condition is reformulated by post-multiplying it by $\mathrm{d}\boldsymbol{x}^*$. Using Eq. (4), the Jacobian of the constraints with
respect to $\boldsymbol{x}$ can be replaced by the Jacobian with respect to $\boldsymbol{c}$:

$$\nabla_x f(\boldsymbol{x}^*, \boldsymbol{c}_r)^T \mathrm{d}\boldsymbol{x}^* = \sum_{i \in \mathcal{I}} \lambda_i^* \nabla_c g_i(\boldsymbol{x}^*, \boldsymbol{c}_r)^T \mathrm{d}\boldsymbol{c}. \tag{7}$$

The expression for $\nabla_x f(\boldsymbol{x}^*, \boldsymbol{c}_r)^T \mathrm{d}\boldsymbol{x}^*$ in Eq. (3) can be replaced by Eq. (7), obtaining the following *first order estimator*:

$$\mathrm{d}z_{\mathrm{est}}^*(\mathrm{d}\boldsymbol{c}) = \nabla_c f(\boldsymbol{x}^*, \boldsymbol{c}_r)^T \mathrm{d}\boldsymbol{c} + \sum_{i \in \mathcal{I}} \lambda_i^* \nabla_c g_i(\boldsymbol{x}^*, \boldsymbol{c}_r)^T \mathrm{d}\boldsymbol{c}. \tag{8}$$

The first term of the estimator represents how the objective function changes with $\mathrm{d}\boldsymbol{c}$ assuming the optimal design $\boldsymbol{x}^*$ does
not change. The second term gives the change in the optimal objective value due to a variation in the constraints, which results
in a change of the optimal design $\boldsymbol{x}^*$. This formulation is based on a first-order differentiation and is valid under the assumption
that the feasible set does not change with $\mathrm{d}\boldsymbol{c}$. Figure 1 illustrates how the two terms of the estimator works.

A pure linear estimator only takes in account the linear variation of the problem with $\mathrm{d}\boldsymbol{c}$ and cannot show the effect of
diminishing returns. Thus we propose an extension of the estimator that captures the non-linear behavior of the constraints,





called *high-order estimator*. By using a higher order expansion instead of a first-order one, and under appropriate assumptions on the objective function and constraints, the following formula is obtained:

$$\mathrm{d}z_{\text{est}}^*(\mathrm{d}\boldsymbol{c}) = \Delta f(\mathrm{d}\boldsymbol{c}) + \sum_{i \in \mathcal{I}} \lambda_i^* \Delta g_i(\mathrm{d}\boldsymbol{c}), \tag{9}$$

where $\Delta g_i(\mathrm{d}\boldsymbol{c}) = g_i(\boldsymbol{x}^*, \boldsymbol{c}_r + \mathrm{d}\boldsymbol{c}) - g_i(\boldsymbol{x}^*, \boldsymbol{c}_r)$, $i \in \mathcal{I}$ and $\Delta f(\mathrm{d}\boldsymbol{c}) = f(\boldsymbol{x}^*, \boldsymbol{c}_r + \mathrm{d}\boldsymbol{c}) - f(\boldsymbol{x}^*, \boldsymbol{c}_r)$. This estimator is valid assuming that (i) the objective function and constraints are linear in $\boldsymbol{x}$ and there is no couplings between $\boldsymbol{x}$ and $\boldsymbol{c}$, (ii) the active set does not change with a finite variation $\mathrm{d}\boldsymbol{c}$, and (iii) constraints that do not depend on $\boldsymbol{c}$ do not affect the change of optimum. The derivation and explanation for the assumptions can be found in Appendix B. Appendix C illustrates how the validity assumptions impacts the performance of the estimator on a simple quadratic program. In addition, Fig. 1 illustrates how the assumptions on the coupling impact the estimator validity.

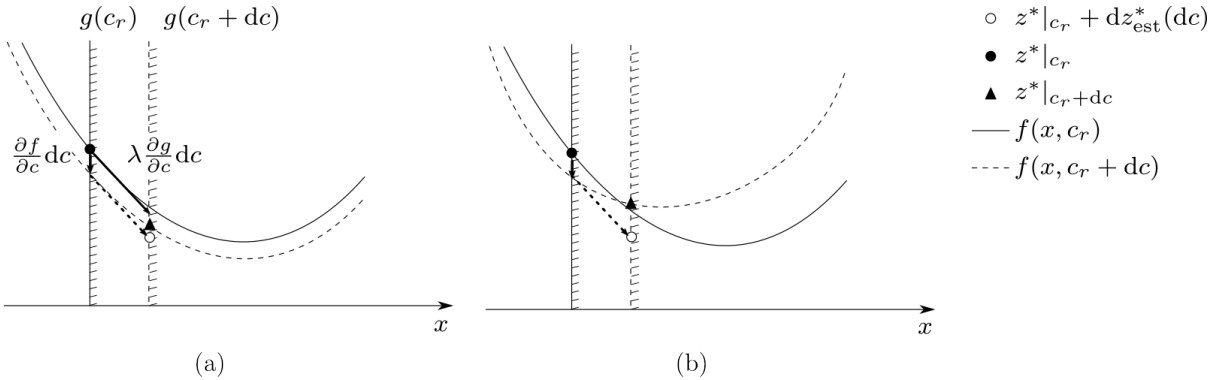

**Figure 1.** Illustration of the estimator on a quadratic problem, with one scalar design variable $x$ and one constraint $g$ represented by the vertical line. The problem is represented for the reference value $\boldsymbol{c}_r$ and in the presence of a variation $\mathrm{d}\boldsymbol{c}$, when the coupling between $x$ and $c$ is weak **(a)** and when it is strong **(b)**. The estimated optimum (white circle) is close to the real optimum (black triangle) only in the weak coupling case.

## 3 Case study

In this section, we present the case study used to evaluate the estimator. We first describe the tower optimization problem on which the estimator is applied. Then, the method to estimate how the optimal tower mass and levelized cost of energy (LCOE) change with the control tuning are described. The third part reports the Linear Quadratic Regulator (LQR) control law and the control tuning used. This section is concluded by describing the analysis and fatigue damage models.

### 3.1 Optimization problem

We consider a wind turbine tower optimization problem with the objective to reduce the LCOE. Two configurations of the tower design are considered: a standard configuration, where the natural frequencies of the structure are required to not interact




with the rotor rotational frequency, and a soft-soft configuration, where the natural frequencies can be lower than the passing frequency and resonance is avoided through active control. In this work, we do not consider the resonance avoidance strategy in the design of the controller. The tower design is parameterized with the tower height $h$, the diameter $\boldsymbol{d}$ and wall thickness $\boldsymbol{t}$ of each tower segment. Geometrical constraints are set on taper, continuity of wall thickness and maximum tower diameter to ensure the tower can be built. The load constraints, $g_{D,j}, j = 1, ..., n_s$ ensure that the fatigue damage does not exceed 1 along the full length of the tower. Finally, for the standard configuration, a frequency constraint is set so that the first and second natural frequencies $f_1, f_2$ are sufficiently far from the rotors 1P frequency $f_{1P}$.

The optimization is represented by Problem 10, where $c = c_r$ represents the scalar control tuning set at its reference value:

$$\underset{h}{\text{minimize}} \quad z = \text{LCOE}(m^*(h, c_r), h)$$

$$\text{with} \quad m^*(h, c_r) = \underset{\boldsymbol{d}, \boldsymbol{t}}{\text{minimize}}\{ m(\boldsymbol{t}, \boldsymbol{d}, h), (\boldsymbol{t}, \boldsymbol{d}) \in \mathcal{S}(h, c_r)\} \tag{10}$$

$$(\boldsymbol{t}, \boldsymbol{d}) \in \mathcal{S}_1(h, c) \leftrightarrow \begin{cases} g_{D_j}(\boldsymbol{d}, \boldsymbol{t}, h, c) \leq 0, & j = 1, ..., n_s \\ f_k(\boldsymbol{x}) \geq \dfrac{f_{1P}}{1 - \delta f}, & k = 1, 2 \\ \text{Geometrical constraints} \end{cases} \tag{11}$$

$$(\boldsymbol{t}, \boldsymbol{d}) \in \mathcal{S}_2(h, c) \leftrightarrow \begin{cases} g_{D_j}(\boldsymbol{d}, \boldsymbol{t}, h, c) \leq 0, & j = 1, ..., n_s \\ \text{Geometrical constraints.} \end{cases} \tag{12}$$

Two sets of constraints $\mathcal{S}_1$ and $\mathcal{S}_2$ expressed by Eq. (11) and (12) are considered, corresponding to the standard and soft-soft configurations, respectively. The tower mass is noted $m$.

The control tuning has a direct impact on the optimization problem through the change in the aerodynamics loads and in the dynamic response of the wind turbine. This in turn impacts the fatigue loads. On the other hand, the AEP used to calculate the LCOE is only marginally impacted by the control tuning, since it is based on the average power production, which tends to be relatively insensitive to such changes.

Problem 10 is formulated using a nested formulation, where the tower mass $m$ is the objective function of the inner optimization problem and acts as an intermediate variable to calculate the LCOE. Solving the equivalent monolithic problem would require excessive computational resources. This is because a large number of aeroelastic simulations is required to accurately estimate the loads, and we use finite-difference to estimate the gradient of the objective function and of the constraints. To avoid this issue, we use a frozen-load approach to reduce the computational cost, under the assumption that the load envelope varies slowly with changes in the inner tower design variables $(\boldsymbol{d}, \boldsymbol{t})$. For a given tower height, a beam model of the tower is derived and integrated into the complete aeroelastic multibody model of the turbine, which is then used to conduct all necessary aeroelastic simulations. The corresponding loads are then *frozen* and used as input for the tower mass optimization. Upon convergence of the inner optimization, the mass difference between the tower design used for the load evaluation and the tower





design found at the end of the optimization is evaluated. If the change in design is greater than a given threshold, the process is repeated iteratively (Bottasso et al., 2016). While this approach can potentially lead to non-optimal design, it is widely used in
wind energy and provides satisfying results.

### 3.2 Estimator applied to the tower mass and LCOE

In the tower optimization problem represented by Problem 10, only the fatigue constraint has a direct dependence on the controller behavior. Therefore, the estimator in Eq. (9) is defined using this constraint only and applied to the tower mass minimization problem. In this case, the tower mass is not a function of the control parameter and the gradient of the objective
function with regards to $c$ is zero. As a result, the change in optimal tower mass $m^*$ is estimated with the following expression:

$$\mathrm{d}m^*_{\mathrm{est}}(\mathrm{d}c) = \sum_{j=1}^{n_s} \lambda_{D,j} \Delta g_{D,j}(\mathrm{d}c), \tag{13}$$

where $\lambda_{D,j}$ represent the Lagrange multipliers of the inner problem associated to the fatigue damage constraint $g_{D,j}$. The validity of the high-order estimator is ensured because the active set is robust, there is little interaction between constraints,
and the objective and constraints tend to be nearly linear around the optimum.

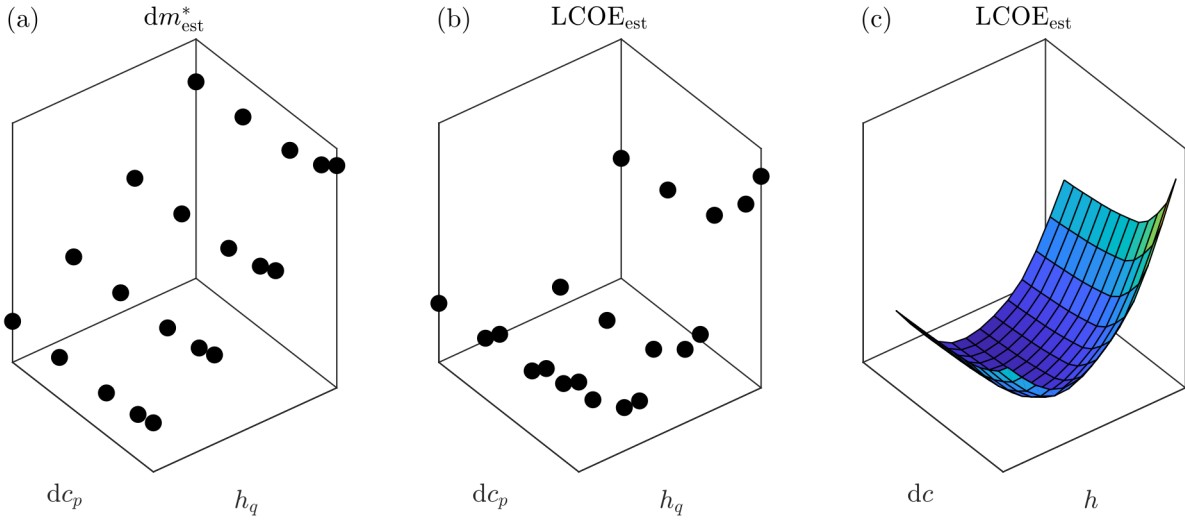

**Figure 2.** Illustration of the process used to make the LCOE estimate function $\mathrm{LCOE}_{\mathrm{est}}$ from the optimal tower mass estimator $\mathrm{d}m^*_{\mathrm{est}}$: the optimal tower mass estimate is obtained over a set of points $\mathrm{d}c_q$ and $h_q$ **(a)**, the corresponding LCOE is calculated using a simplified cost model **(b)** and a quadratic interpolation is run to form the LCOE estimate function **(c)**

The estimator formula cannot be applied directly to LCOE due to the nested formulation of the problem. Instead, we use a surrogate model of the LCOE as a function of the tower mass and tower height. This model is then applied to the optimal tower mass estimator calculated for different tower heights. The process is illustrated in Fig. 2. The resulting LCOE estimate can be





used to gauge the optimal LCOE that would have been obtained by solving the minimization problem including control tuning
as a design variable, i.e. using CCD. This is done by minimizing the LCOE estimate function over the range of data used to
generate the surrogate model, i.e.

$$\text{LCOE}^*_{\text{est}} = \underset{h, \text{d}c}{\text{minimize}} \ \text{LCOE}_{\text{est}}(h, \text{d}c). \tag{14}$$

### 3.3 Control parametrization

We use a wind-scheduled Multi-Input Multi-Output (MIMO) LQR controller with integral action (Bottasso et al., 2012b).
The controller states are the tower top displacement and velocity, the rotational speed, the pitch angle, the pitch rate, and the
electrical torque. The integral of the rotational speed is added to eliminate the steady state error of the controller. The controller
inputs are the pitch angle and the electrical torque. At each wind speed considered in the operational range, the controller gains
are computed by applying LQR theory to the linearized system of the turbine dynamics, see Hendricks et al. (2008) for more
details.

The tuning of an LQR controller is done through the choice of the entries of the weight matrices associated to the states and
inputs, noted $\mathbf{Q}$ and $\mathbf{R}$. In this work, the controller is tuned by changing the diagonal term of $\mathbf{Q}$ associated to the tower top
velocity. The following expression reports the parametrization of the weight matrices:

$$\mathbf{Q}(c) = \begin{bmatrix} 0 & & & & & \\ & c & & & & \\ & & 0 & & & \\ & & & \frac{1}{\beta^2_{\max}} & & \\ & & & & 0 & \\ & & & & & 0 \\ & & & & & & q \end{bmatrix}, \qquad \mathbf{R} = \begin{bmatrix} r & 0 \\ 0 & 0.1 \end{bmatrix}, \tag{15}$$

where $c = 0$ is the nominal control tuning and $\beta_{\max}$ is the maximum pitch angle of the turbine power regulation strategy. A
gain schedule is created by varying the parameters $r$ and $q$ over the operational range.

The choice of parametrization was done by doing a sensitivity analysis of the diagonal entries of the matrices $\mathbf{Q}$ and $\mathbf{R}$ on
fatigue damage, power production, and ultimate loads. The weight matrix entry associated to the tower top velocity was found
to give a good fatigue damage reduction, without affecting the standard deviation of the power production in a significant
manner.

### 3.4 Analysis model

The numerical experiments presented in this work are conducted using the multi-disciplinary wind turbine design optimization
framework `Cp-max`. The details of the framework can be found in the available literature (Bottasso et al., 2012a, 2014, 2016).
We highlight the aspects that are important for tower optimization and fatigue calculations in this section.

The tower is modelled as a steel tubular structure, divided in $n_e$ elements. Each tower element is characterized by its radius
at the top and bottom, and its wall-thickness. The tower is then modelled as a non-linear geometrically exact shear and torsion
deformable beam. This is used in turn in the multi-body model of the wind turbine for the aeroelastic simulations, using the
solver `Cp-Lambda`. The aerodynamics of the wind turbine are modeled using the Blade Element Momentum method.





The fatigue load analysis is performed according to certification standards. Simulations are run from the cut-in to the cut-out wind speed with increments of 2 m.s$^{-1}$. At each considered wind speed, simulations are run for 600s for 6 different turbulent
seeds, excluding the initial transient period. Once the aeroelastic simulations are run, loads are extracted at $n_s$ stations along the tower to compute the stress loading on the structure. A rain-flow counting algorithm is then used on the stress time history to identify the number of loading cycles and their amplitude. Miners rule and the material S-N curve is used to estimate the lifetime fatigue damage at each station (Sutherland, 1999).

## 4 Results

In this section, the estimation method presented in Section 3.2 is applied to re-design of the tower of the IEA 3.4 MW reference onshore wind turbine (Bortolotti et al., 2019). We first study the impact of the control tuning on the fatigue damage constraints. This provides the constraints variation $\Delta g_D$ used in the high-order estimator. Then, we compare the high-order estimator of the optimal tower mass to optimization results. Finally, the tower mass estimator is used to assess how the optimal LCOE would change by using CCD.

All optimization problems are solved using the active set optimization algorithm implemented in the `fmincon` routine of MATLAB (The MathWorks Inc., 2019). The outer optimization is solved with tolerance on the expected objective function change $\epsilon_{\mathrm{obj}} = 1e-5$. The inner optimization is solved with $\epsilon_{\mathrm{obj}} = 1e-4$, and with a tolerance on constraint violation $\epsilon_{\mathrm{con}} = 1e-2$. The objective function for the outer and inner problems are both scaled by the corresponding value of the initial design. The number of tower elements is $n_e = 10$, and the number of fatigue damage constraints is $n_s = 19$.

### 4.1 Control action on the fatigue damage constraint

Fatigue damage is evaluated for different values of the control tuning variation $\mathrm{d}c$ on a reference tower design. This tower design corresponds to the solution of the inner optimization of Problem 10, solved for $c_r = 0$ and for the reference tower height $h_r = 110$ m. Figure 3 shows that on average, varying the control tuning from 0 to 0.3 reduces the fatigue damage by 6.8%. The fatigue damage reduction varies depending on where the fatigue damage constraint is calculated on the tower. In
particular, the control tuning has a marginal impact at the tower top, corresponding to Constraint 19 in Fig. 3.

### 4.2 Estimator performance on the optimal tower mass

In this section, the change in optimal tower mass due to a control tuning variation is estimated using the results of the previous section. The estimator is then compared to the solution of the tower mass optimization problem run for different variations of the control parameter at the reference tower height.

We first look at the importance of the different constraints on the design, by solving the inner tower optimization problem with fixed control tuning $c_r = 0$ and fixed tower height $h_r = 110$ m. Figure 4 reports the optimal design and the Lagrange multipliers for the two considered configurations. For both configurations, the designs are similar. However the presence of the frequency constraints in the standard configuration drives the wall thickness up in the bottom half of the tower. Analysis





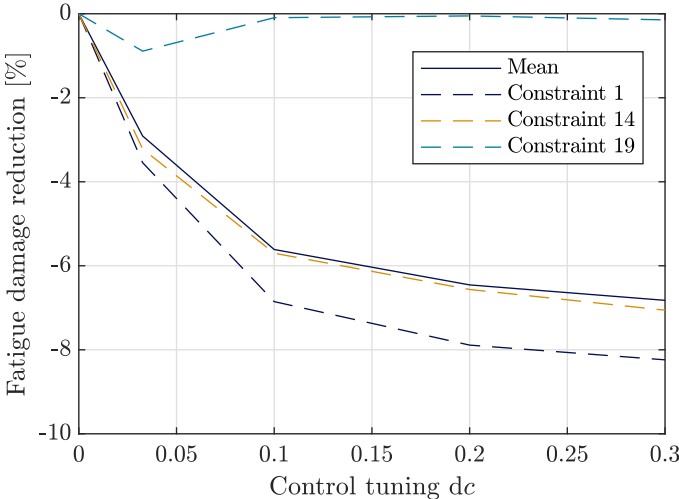

**Figure 3.** Impact of the control tuning on the fatigue damage on average and at three locations along the tower, where Constraint 1 and 19 correspond to the tower bottom and top, respectively.

of the Lagrange multiplier show that for the soft-soft configuration, geometric constraints are the primary drivers. However, these constraints are also insensitive to control tuning. The next most important constraint is fatigue, which can be mitigated by control, indicating potential benefits from CCD. In the standard configuration, the largest Lagrange multiplier is associated with the added frequency constraint, with $\lambda_f = 2.44$. Adding this constraint also reduces the relative importance of fatigue, reducing the potential for CCD, but also showing why the soft-soft tower has lower mass than the standard configuration.

Using the value of the Lagrange multipliers, the first-order and high-order estimators are calculated and reported in Fig. 5. The results of the optimization for $dc = 0.1, 0.2$ and $0.3$ are also reported. The high-order estimator accurately predicts the change in optimal mass for the standard configuration, whereas it under-predicts the results for the soft-soft configuration. Both estimators are able to show that the soft-soft configuration benefits significantly more from a change in control tuning than the standard one, in accordance with the constraint analysis. However, the high-order estimator more precisely quantifies this benefit whereas the first-order estimator fails to capture the effect of diminishing returns on controller tuning.

### 4.3 Estimator performance on the LCOE

In this section, the optimal LCOE is estimated using the results of the previous sections and compared to the results of the control co-design optimization. We want to understand if the LCOE can be reduced by the combined action of control load alleviation and changing the tower height through CCD, and if the proposed estimation method can predict the CCD results.

Figure 6 reports the contour plot of the LCOE estimate function for the standard and soft-soft configurations, calculated as described in Section 3.2 for different tower heights ($0.9h_r, h_r, 1.1h_r, 1.2h_r$) and for $dc = 0, 0.1, 0.2, 0.3$. As expected, there is little coupling between the tower height and the control parameter in the standard configuration, with the LCOE showing only marginal variations with control tuning. For the soft-soft configuration instead, the LCOE can be reduced by simultaneously

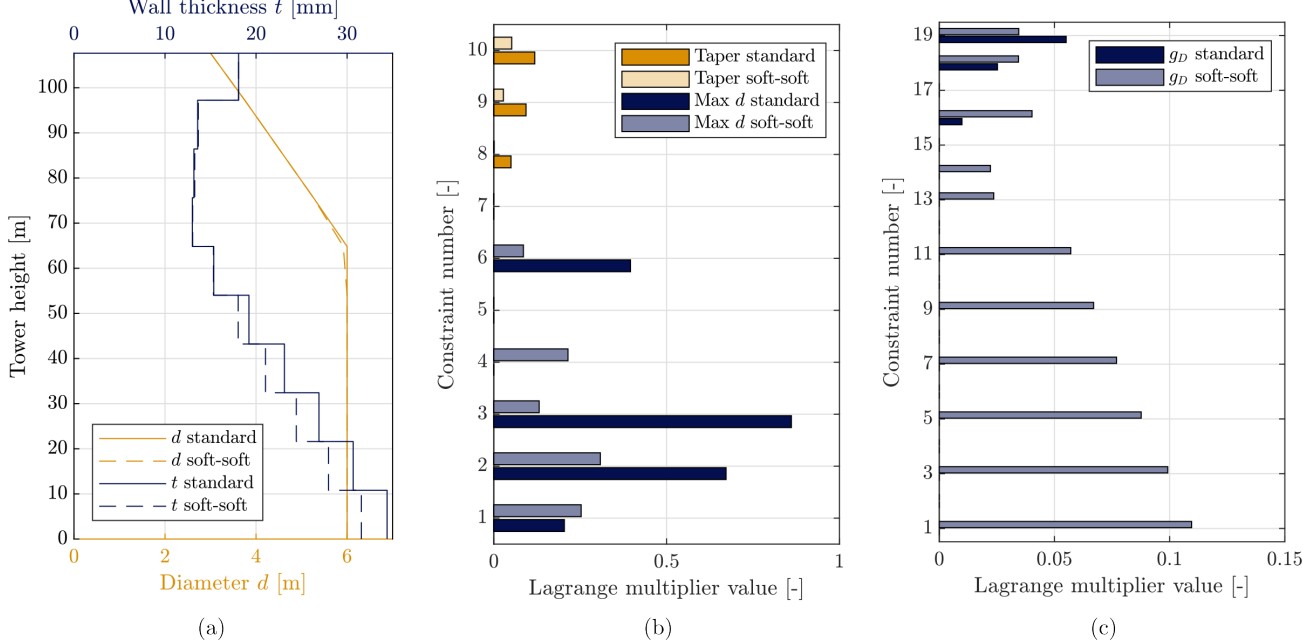

**Figure 4.** Characteristics of the optimal standard and soft-soft tower designs for the reference height $h_r = 110$ m and control tuning $c_r = 0$: optimal tower design **(a)** optimal Lagrange multipliers associated to the fatigue damage **(b)**, and geometric constraints **(c)**.

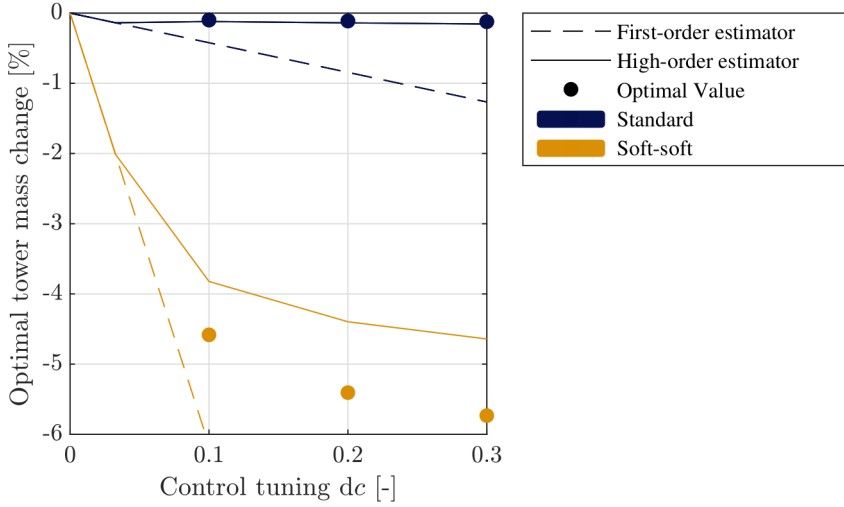

**Figure 5.** Comparison between the optimum mass change $\mathrm{d}m^*$ and the estimated mass change $\mathrm{d}m^*_{\mathrm{est}}$ calculated with the first-order and high-order estimator, for different values of the control parameter and for the two configurations. The tower height is fixed to the reference height.





changing the control parameter and the tower height. The estimated change in optimal LCOE is calculated as the minimum of the estimate function, and marked as a white circle in Fig. 6.

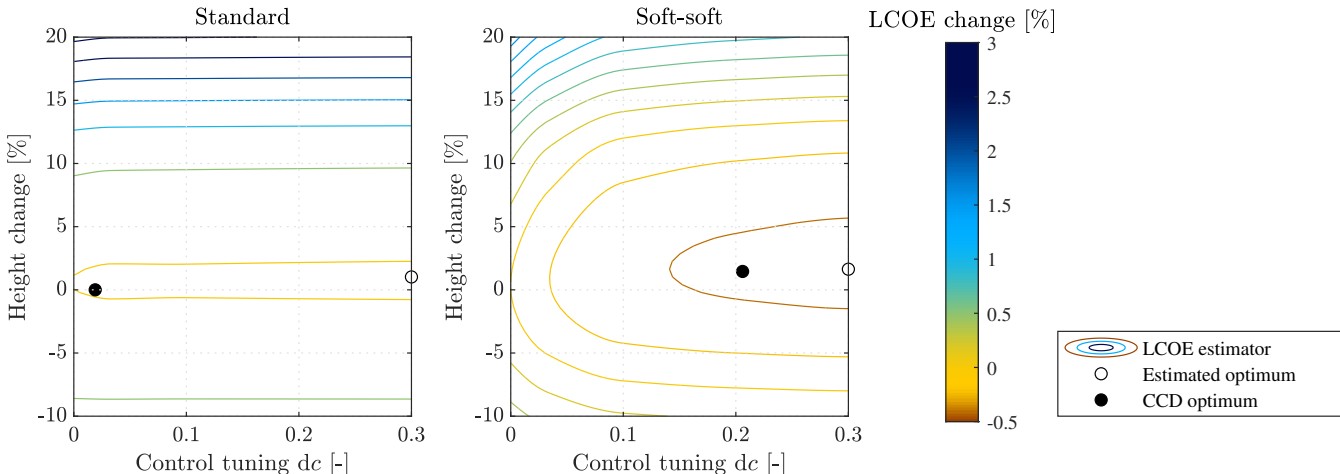

**Figure 6.** Relative change of LCOE as a function of the tower height change and control tuning parameter calculated using the high-order estimator, for the standard and soft-soft configuration. The reference LCOE value is the optimal LCOE for the non CCD problem with $c_r = 0$.

In order to assess the accuracy of the LCOE estimator, we solve the tower optimization problem with a non-CCD formulation (corresponding to Problem 10 with $c_r = 0$) and with a CCD formulation with bounds on the control tuning $c \in [0, 0.3]$. Table 1 reports the change in optimal LCOE brought by the use of CCD calculated directly with the optimization results and with the estimation method. The estimation method correctly predicts that the soft-soft configuration benefits much more from CCD than the standard configuration. In addition, the estimated improvement is accurate compared to the optimization results.

**Table 1.** Change of optimal LCOE between a CCD and a non-CCD approach calculated using the estimation method and using optimization directly.

|  | CCD Optimization | Estimator |
|---|---|---|
| Standard configuration | -0.01% | -0.02% |
| Soft-soft configuration | -0.53% | -0.45% |

In terms of computational cost, calculating the LCOE estimator required solving four tower mass optimization problems and evaluating the fatigue damage for four values of the control tuning, resulting in 12 evaluations of the full set of aero-elastic simulations for each configuration. In comparison, solving the CCD problem required solving the inner problem and running the full set of simulations 50 times for the soft-soft configuration and 20 times for the standard configuration. Therefore, the presented estimation method is able to identify which configuration benefits from a CCD formulation, with a fraction of the computational effort of the actual optimization.





The results of the optimization for the two configurations are reported in Table 2. The data shows that the optimal CCD soft-soft tower is 2.8% lighter and 1.5% higher than the version calculated without CCD, which implies a gain in power capture in sheared inflow. This reduction in tower mass and increase in power capture explains why the LCOE is more impacted for the soft-soft configuration than for the standard configuration. While the estimator performs well on the change in optimal LCOE, it does not predict well the change in design. Indeed, Fig. 6 shows that the estimated change in optimal design is far from the actual one. This is likely caused by the decreasing accuracy of the estimator as $\mathrm{d}c$ increases.

**Table 2.** Comparison of the optimal objective value for the standard and soft-soft configurations, when calculated with and without a CCD formulation. The percentage change between the CCD and the non-CCD cases is reported in parentheses.

|  |  | Standard non-CCD | Standard CCD | Soft-soft non-CCD | Soft-soft CCD |
|---|---|---|---|---|---|
| Tower height $h$ | [m] | 110 | 110.6 (+0.5%) | 110 | 111.6 (+1.5%) |
| Control tuning $c$ | [-] | 0 | 0.019 | 0 | 0.203 |
| Tower mass $m^*$ | [t] | 331.07 | 334.08 (+0.9%) | 311.33 | 302.47 (-2.8%) |
| AEP | [GWh] | 14.955 | 14.977 (+0.1%) | 14.955 | 15.014 (+0.4%) |
| LCOE | [$/Mwh] | 41.481 | 41.477 (-0.01 %) | 41.235 | 41.016 (-0.5%) |

# 5 Discussion

A CCD approach can incur major computational costs when compared to the simpler non-CCD optimization. At the same time, our results show that CCD is not always guaranteed to provide benefits to the final design compared to a more straightforward non-CCD approach. Without knowing a-priori the potential benefit, there is a significant risk, in terms of engineering time, code development and computational resources, in attempting a CCD optimization. This work demonstrates that results from the simplified optimization problem can be used in conjunction with the high-order estimator, to determine whether a given problem can benefit from taking a CCD approach. The first-order estimator shows similar results, however fails to capture the effect of diminishing returns from controller tuning. The method is applicable for similar problems where the optimum design is driven by a load constraint when loads can be alleviated by control action, for example the design of wind turbine support structures or blades. In addition, while the estimation method was developed to target CCD applications, the mathematical derivations and associated assumptions are developed in the general case, where $c$ can be any parameter. Therefore, the method can be applied to any optimization problem to disentangle the effects of one parameter on the rest of the solution.

The validity of the high-order estimator depends on strong assumption on the objective functions and constraints. When the assumptions are violated, the estimator can under-predict the benefits of CCD, as shown in our results. In addition, the estimator uses local sensitivity information of the non-CCD optimum and will be inaccurate when a CCD approach significantly changes the design. Therefore, there may still be a benefit of using a CCD approach, even if the estimator fails to show it.

In this study, we perform CCD using one tuning parameter of the LQR controller. The proposed method is not dependent on the control architecture, but was verified in a case where the controller is tuned using only a few variables. However, CCD





can be performed in several other ways. The applicability of the method to parametrizations with a large number of design variables, for example open-loop control in the context of direct transcription, is left for future work on the topic.

Finally, this work shows how CCD can be used for the design of wind turbine towers. In the presence of an active frequency constraint, CCD may not give significant improvements. Instead, the use of active load alleviation enables a taller and lighter-mass tower compared to the non-CCD design. The control used for the soft-soft configuration did not include an active
resonance avoidance strategy. We can expect that including this feature in the controller design would translate into reduced benefits. In addition, our results are specific to one particular wind turbine and may not be generally applicable. However, these results highlight the importance of doing a thorough analysis of the driving constraints through the use of Lagrange multipliers before attempting to solve a complex and computationally expensive optimization.

## 6    Conclusion

This study shows how design sensitivity analysis can be used to estimate the change of optimal objective value caused by a change in control. Using the solution of an optimization problem with fixed control, we can characterize the results of the more complex control co-design problem without the associated computational effort. Two estimators are presented, based on first-order and high-order approximations, respectively, where the latter captures non-linear effects.

The proposed estimation method is applied to the redesign of a wind turbine tower driven by fatigue loads, using an LQR
controller targeting fatigue load alleviation. High computational resources are required to calculate fatigue damage accurately, which makes this problem an ideal application for the estimator. Two design configurations are considered: a standard configuration, where a frequency constraint is enforced to avoid resonance with the rotational frequency of the rotor, and a soft-soft configuration, where resonance is avoided using active control. The proposed first-order and high-order estimators are applied to the optimal tower mass and optimal LCOE problems. We have shown that the high-order estimator accurately predicts
how the tower mass changes with control tuning, compared to optimization results. The first-order estimator is inaccurate for large values of control tuning, but captures the difference between the standard and soft-soft configurations. Combined with an LCOE surrogate model, the high-order estimator predicts a 0.45% reduction in optimal LCOE for the soft-soft tower, while running the CCD optimization gives an improvement of 0.53%. The proposed estimation method is accurate and uses only a fraction of the computational resources of the CCD optimization. Our results additionally show that the standard tower config-
uration does not benefit from a CCD approach, due to the presence of an active frequency constraint. Changing the control is beneficial for the soft-soft tower, because the fatigue damage constraint is the primary design driver and can be alleviated by control action. In this case, the use of CCD yields a higher tower with lower mass, which impact the LCOE significantly.

As shows in this work, design sensitivity analysis allows to identify relevant design problems for CCD from the results of a simplified non-CCD solution. In a context where computational effort is an obstacle to the wide use of CCD, the proposed
method can help identify and quantify the benefits of this approach for wind energy applications.



## Appendix A: Nomenclature

Symbols used for generic optimization problems

| | |
|---|---|
| $\boldsymbol{\lambda}$ | Lagrange multipliers |
| $\boldsymbol{c}$ or $c$ | Variables or parameters describing the controller |
| $\boldsymbol{c}_r$ or $c_r$ | Reference value for the control variables |
| $f$ | Objective function |
| $g_i, i = 1, ..., n$ | Constraints |
| $\boldsymbol{x}$ | Design variable of the optimization problem, except control |
| $z$ | Objective function value |
| $\mathcal{I}$ | Set of active constraints |
| $\nabla_x \square$ | Jacobian or gradient of $\square$ with regards to $x$ |
| $\square^*$ | Value at the optimum |
| $\mathrm{d}\square$ | Small variation |
| $\mathrm{d}\square_{\mathrm{est}}$ | Estimated value of the variation of $\square$ |

Symbols used for the tower design optimization problem

| | |
|---|---|
| $\lambda_{D,j}, j = 1, ..., n_s$ | Lagrange multipliers associated to the fatigue damage constraint |
| $\lambda_f$ | Lagrange multipliers associated to the first frequency constraint |
| $\boldsymbol{d}$ | Diameter of the tower elements |
| $f_1, f_2$ | First and second natural frequencies of the turbine |
| $f_{1P}$ | Rotor 1P passing frequency |
| $g_{D,j}, j = 1, ..., n_s$ | Fatigue damage constraints |
| $h$ | Tower height |
| $m$ | Mass of the tower |
| $n_e$ | Number of tower elements |
| $n_s$ | Number of fatigue damage constraints |
| $r, q$ | Gain-schedule parameters for the LQR control gains |
| $\boldsymbol{t}$ | Thickness of the tower elements |

Abbreviations

| | |
|---|---|
| AEP | Annual energy production |
| CCD | Control co-design |
| LCOE | Levelized Cost of Energy |
| LQR | Linear quadratic regulator |





**Appendix B: High-order estimator**

In this appendix, we derive the high-order estimator expressed by Eq. (9) and explain the validity assumptions. We consider the following non-linear optimization problem:

$$\underset{\boldsymbol{x}}{\text{minimize}} \quad z = f(\boldsymbol{x}, \boldsymbol{c}_r)$$

$$\text{subject to} \quad g_i(\boldsymbol{x}, \boldsymbol{c}_r) \leq 0 \quad i = 1, ...n. \tag{B1}$$

The change of optimal objective value due to a change of the control parameter $\mathrm{d}\boldsymbol{c}$ is defined as:

$$\mathrm{d}z^*(\mathrm{d}\boldsymbol{c}) = f(\boldsymbol{x}^* + \mathrm{d}\boldsymbol{x}^*, \boldsymbol{c}_r + \mathrm{d}\boldsymbol{c}) - f(\boldsymbol{x}^*, \boldsymbol{c}_r). \tag{B2}$$

We assume that the objective function $f$ is linear in $\boldsymbol{x}$ and that does not admit a coupling between the variables $\boldsymbol{x}$ and $\boldsymbol{c}$. Using these assumptions on a second-order Taylor expansion of Eq. (B2) gives:

$$\mathrm{d}z^*(\mathrm{d}\boldsymbol{c}) = f(\boldsymbol{x}^* + \mathrm{d}\boldsymbol{x}^*, \boldsymbol{c}_r + \mathrm{d}\boldsymbol{c}) - f(\boldsymbol{x}^*, \boldsymbol{c}_r) = \nabla_x f(\boldsymbol{x}^*, \boldsymbol{c}_r)^T \mathrm{d}\boldsymbol{x}^* + \nabla_c f(\boldsymbol{x}^*, \boldsymbol{c}_r)^T \mathrm{d}\boldsymbol{c} + \cancel{\frac{1}{2}\mathrm{d}\boldsymbol{x}^{*T}\nabla_x^2 f(\boldsymbol{x}^*, \boldsymbol{c}_r)\mathrm{d}\boldsymbol{x}^*}$$

$$+ \frac{1}{2}\mathrm{d}\boldsymbol{c}^T \nabla_c^2 f(\boldsymbol{x}^*, \boldsymbol{c}_r)\mathrm{d}\boldsymbol{c}^* + \cancel{\mathrm{d}\boldsymbol{x}^{*T}\nabla_{xc}^2 f(\boldsymbol{x}^*, \boldsymbol{c}_r)\mathrm{d}\boldsymbol{c}} + o(||\mathrm{d}\boldsymbol{c}||^2). \tag{B3}$$

We use the notation $\nabla_x^2 \square$ for the Hessian of a function with respect to $x$. Due to the assumption on $f$, the second-order terms dependent on $\mathrm{d}\boldsymbol{x}^*$ are negligible. The remaining terms dependent on $\mathrm{d}\boldsymbol{c}$ can be identified with the second-order Taylor

expansion of the function $\boldsymbol{c} \mapsto f(\boldsymbol{x}^*, \boldsymbol{c})$ around the point $\boldsymbol{c} = \boldsymbol{c}_r$. Therefore, the expression can be rewritten as:

$$\mathrm{d}z^*(\mathrm{d}\boldsymbol{c}) = \nabla_x f(\boldsymbol{x}^*, \boldsymbol{c}_r)^T \mathrm{d}\boldsymbol{x}^* + \Delta f(\mathrm{d}\boldsymbol{c}) + o(||\mathrm{d}\boldsymbol{c}||^2), \tag{B4}$$

where $\Delta f(\mathrm{d}\boldsymbol{c}) = f(\boldsymbol{x}^*, \boldsymbol{c}_r + \mathrm{d}\boldsymbol{c}) - f(\boldsymbol{x}^*, \boldsymbol{c}_r)$. Applying the same assumption on the constraints gives the following expression:

$$g_i(\boldsymbol{x}^* + \mathrm{d}\boldsymbol{x}^*, \boldsymbol{c}_r + \mathrm{d}\boldsymbol{c}) - g_i(\boldsymbol{x}^*, \boldsymbol{c}_r) = \nabla_x g_i(\boldsymbol{x}^*, \boldsymbol{c}_r)^T \mathrm{d}\boldsymbol{x}^* + \Delta g_i(\mathrm{d}\boldsymbol{c}) + o(||\mathrm{d}\boldsymbol{c}||^2), \quad i = 1, ..., n, \tag{B5}$$

where $\Delta g_i(\mathrm{d}\boldsymbol{c}) = g_i(\boldsymbol{x}^*, \boldsymbol{c}_r + \mathrm{d}\boldsymbol{c}) - g_i(\boldsymbol{x}^*, \boldsymbol{c}_r)$, $i = 1, ..., n$. We consider the set $\mathcal{I}$ of active constraints that depends on $\boldsymbol{c}$.

Assuming that the active set does not change with $\mathrm{d}\boldsymbol{c}$, one has $g_i(\boldsymbol{x}^* + \mathrm{d}\boldsymbol{x}^*, \boldsymbol{c}_r + \mathrm{d}\boldsymbol{c}) = g_i(\boldsymbol{x}^*, \boldsymbol{c}_r) = 0$, $i \in \mathcal{I}$, and therefore:

$$\nabla_x g_i(\boldsymbol{x}^*, \boldsymbol{c}_r)^T \mathrm{d}\boldsymbol{x}^* = -\Delta g_i(\mathrm{d}\boldsymbol{c}) + o(||\mathrm{d}\boldsymbol{c}||^2), \quad i \in \mathcal{I}. \tag{B6}$$

We can relate the gradient of the objective function to the gradient of the constraints using the optimality conditions. We assume that $f$ and $g_i$, $i = 1, ..., n$ are differentiable and that strong duality holds for Problem B1. Then, if $\boldsymbol{x}^*$ is optimal, there is a set of Lagrange multipliers $\boldsymbol{\lambda}^*$ satisfying the Karush-Kuhn-Tucker conditions (Boyd and Vandenberghe, 2004). Among

these, the stationarity condition states:

$$\nabla_x f(\boldsymbol{x}^*, \boldsymbol{c}_r) + (\boldsymbol{\lambda}^*)^T \nabla_x \boldsymbol{g}(\boldsymbol{x}^*, \boldsymbol{c}_r) = \boldsymbol{0}. \tag{B7}$$





The stationarity condition is reformulated by post-multiplying it by $\mathrm{d}\boldsymbol{x}^*$ and by separating constraints in and outside set $\mathcal{I}$:

$$\nabla_x f(\boldsymbol{x}^*, \boldsymbol{c}_r)^T \mathrm{d}\boldsymbol{x}^* = -\sum_{i \in \mathcal{I}} \lambda_i^* \nabla_x g_i(\boldsymbol{x}^*, \boldsymbol{c}_r)^T \mathrm{d}\boldsymbol{x}^* - \sum_{i \notin \mathcal{I}} \lambda_i^* \nabla_x g_i(\boldsymbol{x}^*)^T \mathrm{d}\boldsymbol{x}^*. \tag{B8}$$

The terms corresponding to constraints in set $\mathcal{I}$ can be reformulated using Eq. (B6). In addition, we assume that the constraints

that do not depend on $\boldsymbol{x}$ contribute marginally to the change of optimum. This means that either the corresponding Lagrange multiplier is small, or that the change of design $\mathrm{d}\boldsymbol{x}^*$ does not impact the constraint, i.e. $\mathrm{d}\boldsymbol{x}^*$ is orthogonal to the support to the constraint and $\nabla_x g_i(\boldsymbol{x}^*)^T \mathrm{d}\boldsymbol{x}^* \ll 1$. Following these considerations, Eq. (B8) becomes:

$$\nabla_x f(\boldsymbol{x}^*, \boldsymbol{c}_r)^T \mathrm{d}\boldsymbol{x}^* = \sum_{i \in \mathcal{I}} \lambda_i^* \Delta g_i(\mathrm{d}\boldsymbol{c}) + o(||\mathrm{d}\boldsymbol{c}||^2). \tag{B9}$$

The expression for $\nabla_x f(\boldsymbol{x}^*, \boldsymbol{c}_r)^T \mathrm{d}\boldsymbol{x}^*$ in Eq. (B4) can be replaced by Eq. (B9), which gives the equation for the high-order

estimator:

$$\mathrm{d}z^*(\mathrm{d}\boldsymbol{c}) = \sum_{i \in \mathcal{I}} \lambda_i^* \Delta g_i(\mathrm{d}\boldsymbol{c}) + \Delta f(\mathrm{d}\boldsymbol{c}) + o(||\mathrm{d}\boldsymbol{c}||^2). \tag{B10}$$

The high-order estimator formula is derived here using a second-order Taylor expansion. However, we can repeat the reasoning with an arbitrary high order $k$ of the Taylor expansion, resulting in an expression in $o(||\mathrm{d}\boldsymbol{c}||^k)$ instead of $o(||\mathrm{d}\boldsymbol{c}||^2)$.

**Appendix C: Application to a quadratic program**

In this section, we illustrate how the assumptions associated to the high-order estimator impacts its validity. For this purpose, we study the simple quadratic program below, with $\boldsymbol{x} = [x_1, x_2]^T$:

$$\begin{aligned} \underset{\boldsymbol{x}}{\text{minimize}} \quad & z = \boldsymbol{y}^T \mathbf{P} \boldsymbol{y} + \boldsymbol{q}^T \boldsymbol{y} + z_0 \quad \text{where } \boldsymbol{y} = [\boldsymbol{x}, c]^T \\ \text{subject to} \quad & \mathbf{G}\boldsymbol{x} \le g_2 c^2 + g_1 c + g_0 \\ & \mathbf{H}\boldsymbol{x} \le h_0 \end{aligned} \tag{C1}$$

The value of $\mathbf{P}$, $\boldsymbol{q}$, $\boldsymbol{G}$, $g_i$, $i = 0,..2$, $\mathbf{H}$ and $h_0$ can be adjusted to create problems that satisfy or violate the validity assumption for the estimator. The parameter $z_0$ is set so that the optimal objective value of the reference problem is $z^* = 0$. For each type

of problem, we study how the optimum and the estimator $\mathrm{d}z_{\text{est}}^*$ change with the value of $\mathrm{d}c$. The reference problem is always taken for $c = 0$, and $\mathrm{d}c$ varies between 0 and 1.





## C1 The objective function is linear in $x$

In order to represent problems with objective functions linear or non-linear in $x$, the diagonal terms of the matrix $\mathbf{P}$ are varied with a parameter $b$. We use the following:

$$\mathbf{P} = \begin{bmatrix} b & 0 & 0 \\ 0 & b & 0 \\ 0 & 0 & 0 \end{bmatrix}, \ \boldsymbol{q} = \begin{bmatrix} -10 \\ 1 \\ 0 \end{bmatrix}, \ \mathbf{G} = \begin{bmatrix} 1 & 0 \end{bmatrix}, \ g_2 = -4, \ g_1 = 3, \ g_0 = 1, \ \mathbf{H} = \mathbf{0}, \ h_0 = 0 \qquad \text{(C2)}$$

When $b = 0$, the objective function is strictly linear in $x$. With increasing values of $b$, the non-linear terms in the objective function dominate more and more the linear term. We study how the estimator performs for $b = 20, 5$ and $0.1$. For this problem, the objective function is not dependent on $c$.

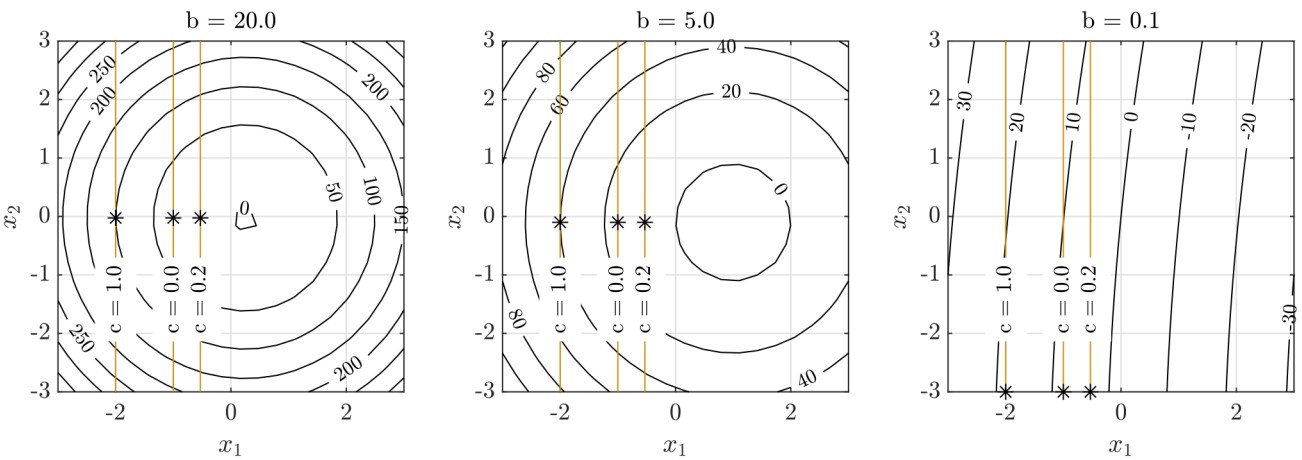

**Figure C1.** Contour plot of the objective function with the optimal value marked with an asterisk (*), for objective functions with varying degree of non-linearity in $x$. The higher the value of $b$, the more dominant the non-linear terms compared to the linear terms in the objective function. The constraint is represented as a yellow line and varies with $c$.

Figure C1 shows the value of the objective as a function of $x_1$ and $x_2$. The constraint $\mathbf{G}x \le g_2 c^2 + g_1 c + g_0$ is represented for different values of $c$ as a yellow line and the optimum is marked as an asterisk. The figure shows that the optimal design changes in a similar way for the different values of $b$. Figure C2 reports the value of the optimum change $dz^*$ and of the first-order and high-order-estimator $dz^*_{\text{est}}$ for the different values of $b$. For low values of $b$ when the objective function is mostly linear in $x$, the high-order estimator follows more closely the optimal value. In addition, we observe that the first-order estimator follows the slope of the optimal value at $c = 0$. This indicates which problems see the most change in optimal value when $c$ is varied.





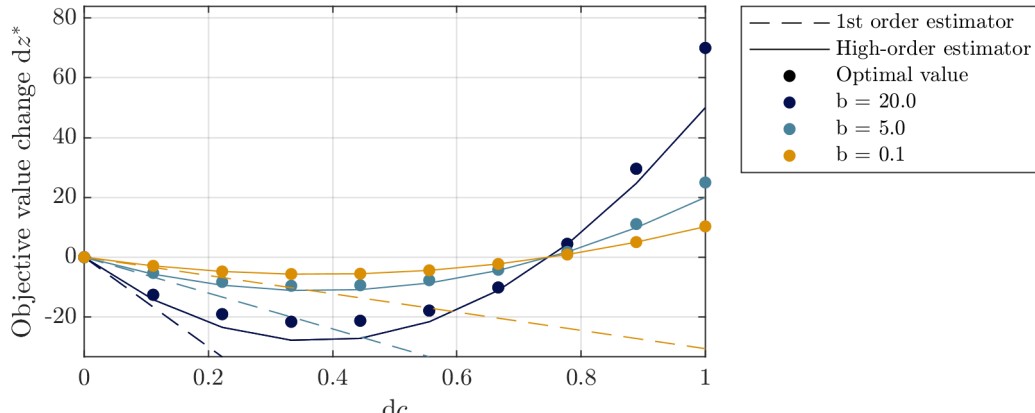

**Figure C2.** Comparison of the optimal objective value with the first-order estimator and the high-order estimator for objective functions with varying degree of non-linearity in $\boldsymbol{x}$. The higher the value of $b$, the more dominant the non-linear terms compared to the linear terms in the objective function.

## C2 There is no coupling between $x$ and $c$ in the objective function

In order to represent the coupling between $\boldsymbol{x}$ and $c$ in the objective function, the non-diagonal terms of the matrix $\mathbf{P}$ corresponding to $x_2$ and $c$ are set to $-b$. We use the following:

$$\mathbf{P} = \begin{bmatrix} 0.1 & 0 & 0 \\ 0 & 0.1 & -b \\ 0 & -b & 0 \end{bmatrix}, \ \boldsymbol{q} = \begin{bmatrix} -10 \\ 0 \\ 0 \end{bmatrix}, \ \mathbf{G} = \begin{bmatrix} 1 & 0 \end{bmatrix}, \ g_2 = -5, \ g_1 = 6, \ g_0 = 1, \ \mathbf{H} = \mathbf{0}, \ h_0 = 0. \tag{C3}$$

The problem is solved for $b = 10.0$, $5.0$ and $0.1$. The higher $b$, the stronger the coupling between $x_2$ and $c$. Figure C3
shows the objective value as a function of $x_1$ and $x_2$ as well as the constraint value for $c = 0.1$ and for $c = 0.2$. The higher the coupling, the larger the changes in the objective function. Figure C4 shows that the estimator performs well only in the case of $b = 0.1$, where the coupling terms are small. Note that in this case, the first-order and high-order estimators do not change with parameter $b$, since they assume that the coupling term is negligible, i.e. $b = 0$.

### C3 The active set does not change with changes in $c$

To study how a change in the active set impacts the validity of the estimator, a constraint is added so that it is not active for $c = 0$ and becomes active as $c$ increases. We use the following:

$$\mathbf{P} = \begin{bmatrix} 0.1 & 0 & 0 \\ 0 & 0.1 & 0 \\ 0 & 0 & 0 \end{bmatrix}, \ \boldsymbol{q} = \begin{bmatrix} -5 \\ 5 \\ 0 \end{bmatrix}, \ \mathbf{G} = \begin{bmatrix} 1 & 0 \end{bmatrix}, \ g_2 = -5, \ g_1 = 6, \ g_0 = 1, \ \mathbf{H} = [1,0], \ h_0 = 0. \tag{C4}$$

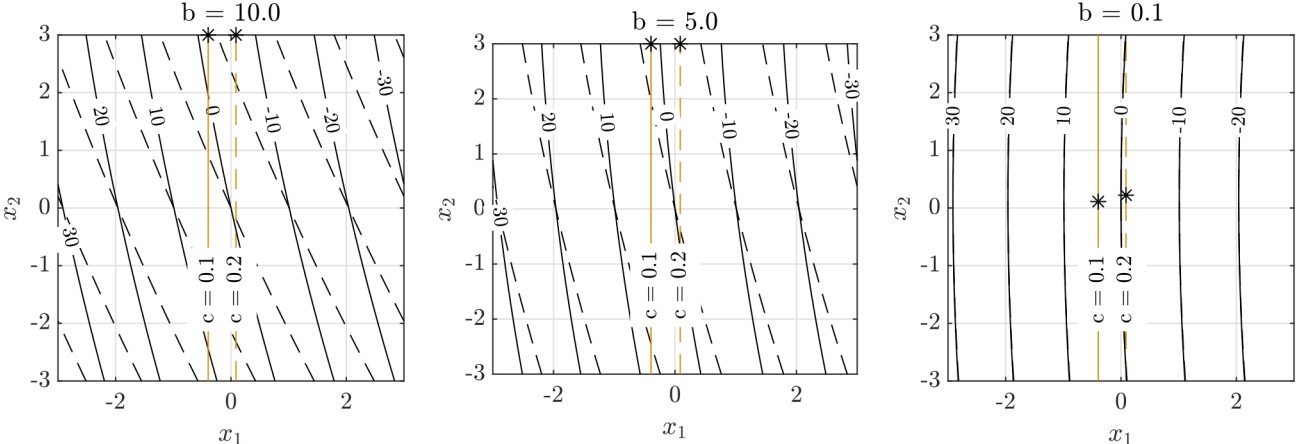

**Figure C3.** Contour plot of the objective function with the optimal value marked with an asterisk (*), for problems with varying degree of coupling between $x$ and $c$ in the objective function. The higher $b$, the more dominant the coupling terms compared to the linear terms in the objective function. Results are represented with a solid line for $c = 0.1$, and with a dashed line for $c = 0.2$ in order to highlight the magnitude of the coupling between $x$ and $c$.

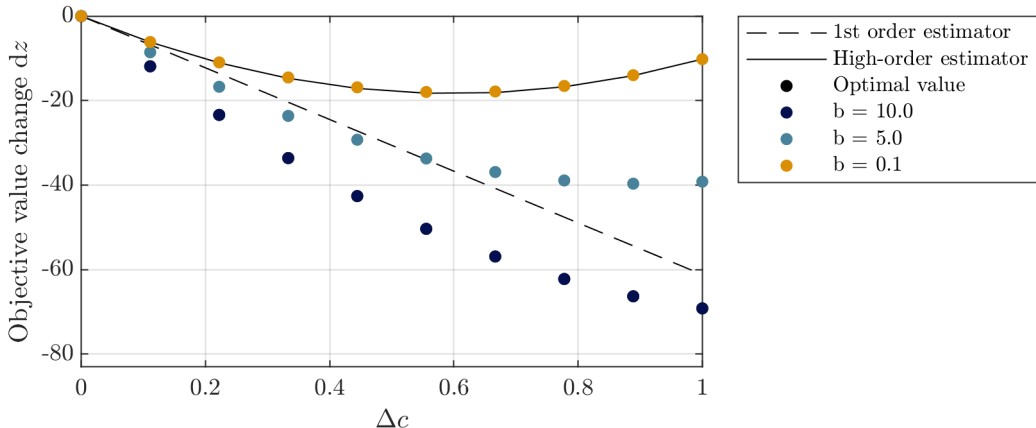

**Figure C4.** Comparison of the optimal objective value with the first-order estimator and the high-order estimator, for problems with varying degree of coupling between $x$ and $c$ in the objective function. The higher $b$, the more dominant the coupling terms compared to the linear terms in the objective function. The high-order estimator assumes $b = 0$.



Figure C5 a reports the objective function with the constraint $\mathbf{G}x \leq g_2 c^2 + g_1 c + g_0$ in yellow and the constraint $\mathbf{H}x \leq h_0$ in blue. For $c = 0$ and $c = 0.1$, the yellow constraint is active. However, for $c = 0.7$, the yellow constraint is no longer active

and the blue constraint becomes active. Therefore, the optimum is set where the blue constraint is, and not where the yellow constraint is. In the region where the active set changes ($c > 0.2$), the high-order estimator does not follow the optimal value anymore.

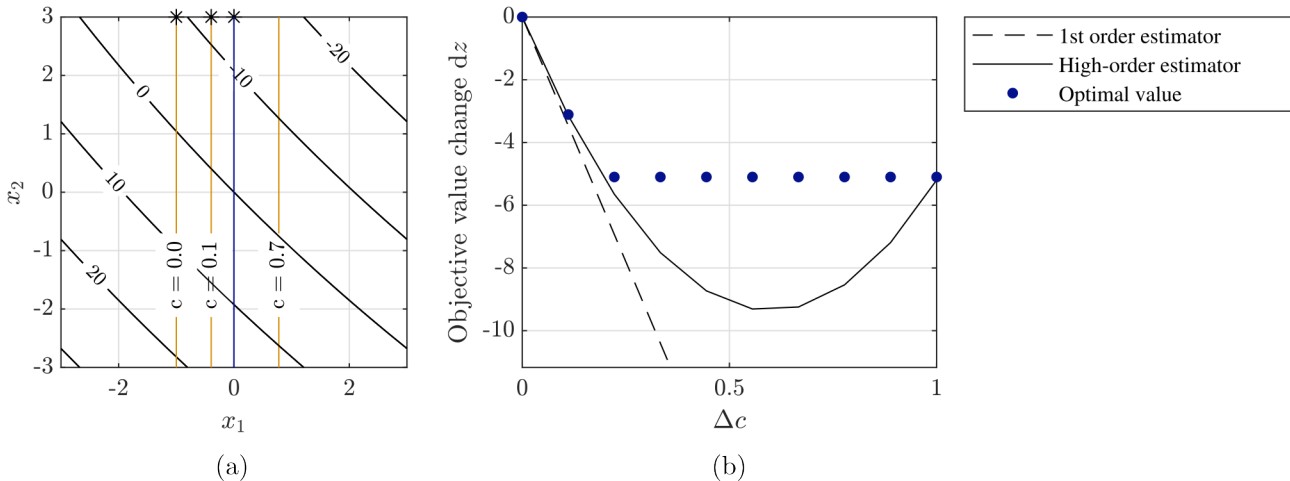

**Figure C5.** Contour plot of the objective function with the optimal value marked with an asterisk (*), where the blue line represent the constraint non-dependent on $c$ **(a)**. Comparison between the first-order, the high-order estimator and the optimal objective value for variations in $c$ **(b)**.

### C4 The constraints non-dependent on $c$ have a small impact on the optimum

In this case study, the constraint non-dependent on $c$ are modeled as $x_1 - bx_2 \leq 0$. We use the following:

$$\mathbf{P} = \begin{bmatrix} 0.1 & 0 & 0 \\ 0 & 0.1 & 0 \\ 0 & 0 & 0 \end{bmatrix}, \; \boldsymbol{q} = \begin{bmatrix} -10 \\ 1 \\ 0 \end{bmatrix}, \; \mathbf{G} = \begin{bmatrix} 1 & 0 \end{bmatrix}, \; g_2 = -5, \; g_1 = 6, \; g_0 = 1, \; \mathbf{H} = [1, -b] \; h_0 = 0 \quad \text{(C5)}$$

Figure C6 reports the objective value and constraints for $b = 0.3$, 1.0 and 100. For $b = 100$, the constraint $x_1 - bx_2 \leq 0$ in blue interacts weakly with the yellow constraint that depends on $c$. This represents a case where the constraint have a small impact on the objective value. For lower values of $b$, we observe that the optimum moves in a different direction than the change in the yellow constraint. This indicates that the yellow and blue constraints are coupled more strongly, and the change

in optimum cannot be attributed mainly to the alleviation of the yellow constraint. Figure C7 shows how the optimal objective value changes in comparison to the estimator. For cases where the two constraints interact weakly ($b = 100$), the estimator follows closely the change in optimal objective value.

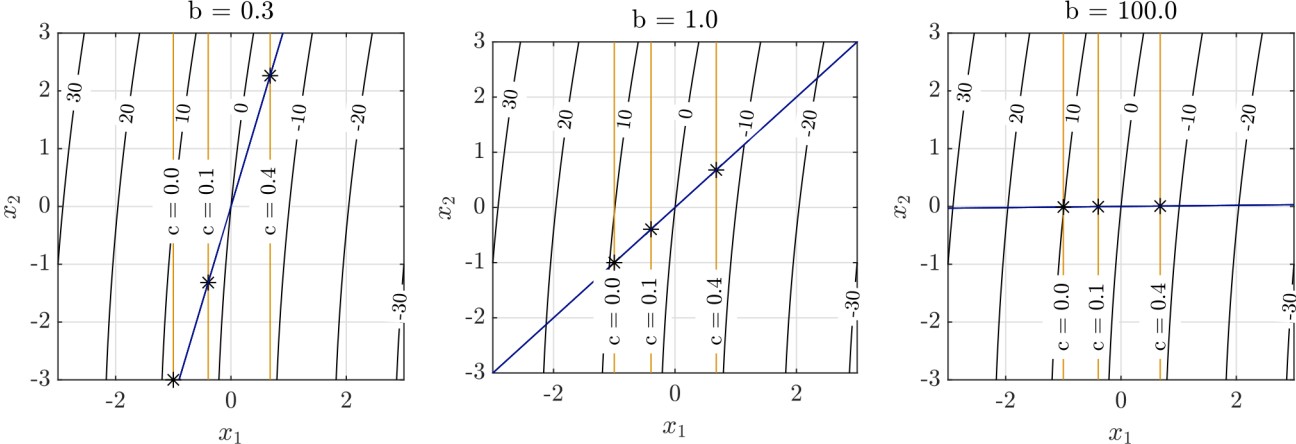

**Figure C6.** Contour plot of the objective function with the optimal value marked with an asterisk (*) for problems where the constraint non-dependent on $c$ (in blue) interacts to a varying degree with the constraint dependent on $c$ (in yellow). The higher the value of $b$, the weaker the interaction with the two types of constraints.

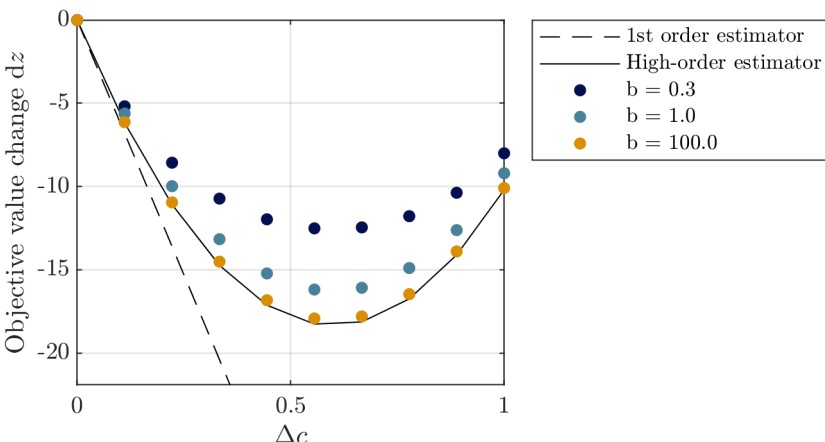

**Figure C7.** Comparison of the optimal objective value with the first-order estimator and the high-order estimator for problems where the constraint non-dependent on $c$ interacts to a varying degree with the constraint dependent on $c$. The higher the value of $b$, the weaker the interaction with the two types of constraints.



*Author contributions.* JI developped the proposed method and implemented the numerical experiments in Cp-max. CLB supervised the research. JI wrote the paper, with inputs from CLB and MKM. All authors provided important input to this research work through discussions,
feedback and by writing the paper.

*Competing interests.* CLB is a member of the editorial board of Wind Energy Science. The authors have no other competing interests to declare.

*Acknowledgements.* This work was funded by the Technical University of Denmark through the PhD project "Multi-disciplinary Design Optimization of Wind Turbines with Smart Blade Technology". It was conducted during an external research stay at the Chair of Wind
Energy of the Technical University of Munich. The authors would like to acknowledge H. Doruk Aktan and Helena Canet at the Technical University of Munich for their valuable help with Cp-max. In addition, the authors would like to thank Mathias Stolpe for his valuable feedback during this work.



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
