# Peer review of "How to identify design optimization problems that can be improved with a control co-design approach?"

_Wind Energy Science, 2023_

## Author Comment (AC1)

**Reply to Reviewers**

We thank the reviewers for their detailed analysis and constructive input. A list of point-by-point replies to the reviewers' comments is detailed in the following.

Additionally, we have taken the opportunity of this revision to make several small editorial changes to the text, in order to improve readability.

A revised version of the manuscript is attached to the present reply, with the main changes highlighted in red (deletions) and blue (additions).

Best regards.
The authors

**Reviewer 1**
**Overall Thoughts**

*Overall, this manuscript has a clear objective and application. The authors propose a method to estimate the impact of changes in control design variable on the objective function in an optimization study without directly solving a control co-design (CCD) optimization problem. The authors argue that this method would save the user time in identifying a scenario where the objective function is insensitive to the control design variable—making CCD somewhat trivial—without actually going through the effort of solving the more complex problem. This intention to reduce computational time in the design process is well-motivated, but I believe the discussion of this point could be more comprehensive.*

*A second thread of the manuscript is the application of the proposed method to the optimal design of a wind turbine tower with fatigue load constraints. I believe the details of this case study convolute and overshadow the main objective at times, and a more refined description is necessary.*

*After addressing these comments and some other minor revisions, I believe this manuscript will be suitable for publication in Wind Energy Science.*

**Major Comments**

- *Line 171: What is the surrogate model for LCOE and how was it calibrated? The model form of the objective function and its dependence on the design variables is necessary information for the reader to understand the case study.*
  **Authors:** We agree with this remark. In the revised manuscript, the objective function is reported in Section 3.3 and the estimate for the cost of energy is described more precisely in Section 4. While revising the manuscript, we found an inconsistency in the text: the objective function was reported as the Levelized Cost of Energy (LCOE), whereas it was in fact the Cost of Energy (CoE). This error does not impact the results or conclusion of the study.

- *Line 189: "A gain schedule is created by varying the parameters r and q over the operational range." I don't think this phrase is intuitive for a reader unfamiliar with LQR control. Also, a few lines later: "The weight matrix entry associated to the tower top velocity was found to give a good*

*fatigue damage reduction, without affecting the standard deviation of the power production in a significant manner."* This description is vague and not reproducible. Similar to the previous comment, the details of the case study (LQR control law, LCOE minimization, etc.) are extensive compared to the objective function sensitivity estimator. I think they require more explanation to be comprehensible by the reader or should be removed or moved to an appendix.

**Authors:** The values of the parameters r and q have been added to Equation (13) in order to ensure reproducibility of the results. The paragraph related to the choice of parametrization has been removed.

- *Line 261: I think the comparison of computational cost between the proposed estimator and the full CCD solution could be clearer. The authors state that "12 [shouldn't this be 16?] evaluations of the full set of aeroelastic simulations for each configuration" were run for the estimator, while the CCD problem requires "running the full set of simulations 50 times for the soft-soft configuration and 20 times for the standard configuration." I'm not sure why these specific numbers of simulations were chosen. Furthermore, how could this result generalize outside of the tower design case study?*

  **Authors:** A section describing the computational effort has been added (Section 5.3) with more details. We have chosen to use the number of evaluations of the full set of aeroelastic simulations as the main metric for the computational effort since it is the most computationally expensive step in the analysis. In addition, the wall time for the entire process depends heavily on the hardware (HPC or workstation) or the aero-elastic solver used. Nonetheless, we report the wall time normalized by the CCD value as a secondary metric.

  The number of full set evaluations for the estimator is now explained in the text:

  > *(l. 292-296) "The fatigue damage constraints are evaluated for four different values of the control tuning, and require one full-set evaluation each. The Lagrange multipliers are evaluated for four different tower heights, and require between one and two full-set evaluations each, depending on the number of iterations in the frozen-load loop. As a result, the estimator is calculated using a total of 11 or 12 full-set evaluations depending on the configuration."*

  We expect similar computational effort benefits for comparable structural optimization problems, where the fatigue damage constraint is driving. Instead, if the driving constraint is easier to calculate, the computational cost for the estimation method will likely drop. The discussion section has been updated to reflect this aspect:

  > *(l. 312-316)"The method is applicable to similar problems where the optimum design is driven by a load constraint, when loads can be alleviated by control action (for example, the design of wind turbine support structures or blades). The computational cost reduction should be similar in problems where the fatigue damage constraints are driving the design. In cases where the driving constraints are easier to evaluate, there should be a greater reduction in computational effort, since the estimator would be less expensive to compute."*

**Minor Comments**

- *Title: I find the question format of the title to be a bit strange and not descriptive of the particular methods introduced by the paper. Perhaps something in the vein of: "Identifying design*

*optimization problems for control co-design approach with objective function sensitivity estimator"?*

**Authors:** We agree that the title should be more descriptive, and have updated it following this comment and a related comment from Reviewer 2: *"A sensitivity-based estimation method for investigating control co-design relevance"*

- *Line 95: How valid is the simplification to only consider active constraints, and assuming that the active set of constraints does not change with control tuning? If your constraint is a function of control tuning, it seems plausible that a change in control could cause an inactive constraint to become active.*

  **Authors:** The assumption that the active set does not change is important for the precision of the estimator. This formula cannot capture the effects of an inactive constraint becoming active with a change of control. We address this aspect in the discussion

  > (l. 319-323) *"The precision of the high-order estimator depends on several assumptions on the objective functions and constraints. When the assumptions are violated, the estimator can under-predict the benefits of CCD, as shown in our results. In addition, the estimator uses local sensitivity information of the non-CCD optimum, and therefore it will be inaccurate when a CCD approach significantly changes the design. Consequently, there may still be a benefit of using a CCD approach, even if the estimator fails to show it."*

- *Line 114: There is repeated use of the phrase "diminishing returns" of controller tuning without any elaboration on what the authors mean. I understand that in Figure 5, a marginal increase in control tuning leads to a diminishing change in optimal tower mass. I think this point could be clearly defined earlier in the manuscript if this phrase is to be used.*

  **Authors:** By "diminishing returns", we mean a non-linear impact of the control tuning on the constraints and objective function. This point has been clarified: (l. 113-114)*" A purely linear estimator only takes into account the linear variation of the problem with d$c$ and cannot capture non-linear effects such as diminishing returns."*

- *Line 133: Is resonance avoidance not included in the soft-soft configuration in order to simplify the problem and focus on the objective function sensitivity? This is a valid approach, but I think the authors should state that point clearly. Otherwise, the problem definition for the soft-soft case seems impractical.*

  **Authors:** We agree with this comment. The following sentence has been added in the paragraph describing the optimization problem: (l. 141-144) *"In this work, the controller design of the soft-soft configuration is kept simple in order to focus on the objective function sensitivity. We assume that the controller is designed in such a way as to operate immediately below and above the resonant frequency, using a classical frequency skipping approach (Bossanyi, 2000). However, for simplicity, we did not implement this feature in the controller, and we simply avoided running simulations in proximity of the resonant condition."*

- *Line 147: "On the other hand, the AEP used to calculate the LCOE is only marginally impacted by the control tuning, since it is based on the average power production, which tends to be relatively*

*insensitive to such changes." Is there a source to justify this statement?*

**Authors:** The annual energy production is calculated from the average power production. Instead, the goal of the controller tuning is to maintain the power production at its desired level despite perturbations or turbulence. So, the AEP should be relatively insensitive to the controller tuning, provided that an adequate controller is chosen. Nonetheless, the sentence has been updated for clarification: (l. 222-223)*" However, the net annual energy production is mostly driven by the tower height, whereas the impact of the controller tuning and the inner tower design is marginal in comparison"*

- *Line 149: I have trouble following the process for solving the optimization problems. For a given tower height, the loads on the tower are simulated and used to optimize the tower mass. When is the height of the tower changed and LCOE minimized? It seems like the solution of the outer optimization problem has not been described. Also, it is stated that "If the change in [tower mass] design is greater than a given threshold, the process is repeated iteratively (Bottasso et al., 2016)." What threshold is used in this paper?*

  **Authors:** It is outside the scope of this paper to describe the details of the optimization framework Cp-max. This framework is documented in several research papers. The threshold used for the frozen-load loop is 1% on the change of tower mass, as now noted in the revised manuscript.

- *Line 203: I think more details on the aeroelastic simulations could be given. What certification standards were used for fatigue analysis? How is turbulence synthetically generated and modeled? How long is the transient period?*

  **Authors:** We provide in the manuscript a reference to Bottasso et al. (2016) where information on the aero-elastic simulations can be found. In addition, we have added a reference to the IEC 64200 standard in the text. The transient period is 30 seconds, and the turbulence is generated using TurbSim (Jonkman and Kilcher, 2012). The text has been updated accordingly.

- *Line 203: How is the fatigue damage resulting from different wind speeds combined into a single estimate of lifetime fatigue damage? Is there a probability distribution of wind speeds that inform taking a weighted average of the different fatigue values?*

  **Authors:** It is outside the scope of this paper to describe in detail the fatigue damage calculation. We provide at the beginning of Section 3.3 a reference to Bottasso et al. (2012) where the details of the structural model can be found.

- *Figure 4: Panel 'b' contains the Lagrange multipliers for geometric constraints, and 'c' for the fatigue damage. The caption appears to have the wrong labels.*

  **Authors:** The caption has been corrected.

- *Table 1: The caption could be simpler and less confusing. The reference for the table is the optimization solution without CCD for a control input of zero. Then, LCOE is optimized with CCD*

*and with the estimator method and compared relative to that reference value. There are a few points in the manuscript where the change of optimal LCOE is presented without a clear definition of what the reference value is.*

**Authors:** Thank you for raising this issue. The header of Table 1 has been modified to remove "CCD", and the caption has been modified as follows:

> *"Table 1: Percentage improvement on the optimal CoE using a CCD approach, calculated with optimization results and the estimation method".*

We want to clarify that Table 2 is included in order to document the optimization results used to produce the data in Table 1. The text introducing this table has been adjusted for this purpose in the updated manuscript.

- *Line 269: The authors state that the estimator accurately predicts the optimal objective value, but not the optimal design. I think it could be useful to clarify here that the goal of this method is to identify how much the objective function can be improved by control tuning, and in cases where the estimator signals much potential, a full CCD study would be performed. Otherwise, the reader could jump to the conclusion that the proposed method would not ultimately reach an optimal design.*

  **Authors**: We agree with this comment. The sentence referred to by Reviewer 1 has been updated as follows:

  > (l. 279-281) *"We note that the estimated change in optimal design is far from the actual one in Fig.5. This is coherent with the goal of the presented method to quantify the sensitivity of the optimal objective value and not of the optimum."*

- *Line 284: Would there be any advantage to quantifying uncertainty of the estimator? In other words, high uncertainty in the estimator could encourage a user to explore the CCD problem regardless of the compared sensitivity in the objective function.*

  **Authors:** There would certainly be an advantage to studying the uncertainty associated with the proposed method. Such work would require the study of uncertainty quantification applied to design sensitivity analysis and could be the topic of future work. However, the relevance of such a study could be limited since CCD optimization is not a widely used approach.

**Typographical Comments**

- *Equation 10: Could be split up into two separately numbered equations, one for the "outer" optimization problem and one for the "inner" problem.*

  **Authors:** We have considered splitting Equation 10 as you suggested, but have opted to keep it as is for conciseness.

**Reviewer 2**

*The paper considers control co-design (CCD) for wind energy systems, with as specific case study, the design of soft-stiff and soft-soft wind turbine towers. Such integrated design of the system and its controller is*

*typically computationally demanding, due to the computational cost of an analysis for each controller design. Specifically for tower design, load calculations determine this cost.*

*The paper presents a methodology to create a simplified version of the CCD optimization problem that is less computationally demanding, but still can provide information about the interaction between system and control design. The idea is that first solving the simplified optimization problem, its results can make it clear whether or not it is worth it to solve the CCD optimization problem.*

*The paper proposes approximations ('estimators') for the CCD problem. They are based on decoupling the design of the system and the controller, assuming the latter fixed. So for a fixed controller design, an optimal system design x\* is obtained. The CCD problem is then approximated by doing first-order and second-order approximations of the neighborhood of x\* as a function of controller parameters.*

*When applied to the tower design case studies, it is seen that the first-order gives some, but arguably too limited information about the usefulness of doing CCD. The second-order approximation does give sufficient information. The results are that the effort of doing CCD is worth it for soft-soft towers, but not for soft-stiff towers.*

**General comments**

*The paper discusses a topic of widespread interest in the wind energy systems design community: approaches to reduce the computational burden of design optimization. Any advances in this area are scientifically relevant. The paper discusses existing literature touching on CCD-type approaches to show specific interest in this area.*

*The approximations proposed as part of the methodology are theoretically nontrivial. They require careful derivation of gradients and higher-order derivatives, adding assumptions to simplify expressions obtained. A good part of the paper is dedicated to this, including two appendices, one of which contains a very commendable and informative analysis of how their approximations can break. The notation is generally good, but would need to be introduced more carefully in advance, to avoid readers having to deal with too much while trying to understand the derivations. Furthermore, the assumptions made at different locations in the paper should be made more explicit, to make sure readers have a clear view of the approximation's limitations.*

*These theoretical discussions are performed on an abstract formulation of the CCD optimization problem. The case study's concrete optimization problems are not directly formulated as such. There is also not a clear translation of this concrete problem to the abstract one, to the detriment of the reader's understanding. What complicates matters is that the simplification performed is not limited to just the approximations introduced, but also involves a surrogate model for one aspect of the concrete optimization problem. With the current presentation, it cannot be expected that readers can understand how the concrete and abstract problems are related with reasonable effort.*

*The goal of the methodology is, effectively, to substantially reduce the computational cost associated to the optimization of a wind energy system. Therefore, it is as important to get a good quantitative view of computational cost (or time, given fixed computational resources) next to the accuracy of the approximations. In the paper, the accuracy is sufficiently described, but the computational cost is not. It is dealt with in one paragraph, which is unclear and in one possible way of reading it may even imply that there is not much difference between solving the simplified problem and the full CCD problem. (In terms of*

*costly load calculations: 50 vs. 16 for soft-soft and 20 vs. 16 for soft-stiff.) Were this reading to be correct, this would severely undermine the significance of this paper.*

- **Authors:** In order to highlight the benefits in terms of computational effort, a dedicated section has been added (Section 5.3). We have included a table comparing the number of costly load evaluations, the number of iterations of the outer-optimization, as well as the wall time relative to the CCD optimization. In the case of the standard tower configuration, the number of full set evaluations is 20 for the CCD optimization and 11 for the estimator. In this case, the difference in computational effort is small. This is likely because the initial design is very close to the optimal design, and the optimization algorithm requires only 4 iterations to converge.  We have added the following sentence at the end of Section 5.3 to clarify this point:

  (l. 298-301) *"We note that the number of iterations for the outer optimization for the two CCD cases is low. For more complex problems or using a tighter optimization tolerance, the number of iterations is likely to increase significantly, and the computational effort of the CCD process will also increase."*

  Furthermore, we would like to highlight that the estimation method allows us to understand the reason why a CCD approach would be beneficial or not. In the submitted manuscript, this point was not clear, so we have added the following sentence in the discussion section (Section 6):

  (l. 308-311) *"Furthermore, the analysis of the Lagrange multipliers and constraint sensitivity in the proposed method gives a justification for why a CCD approach would fail. This information is generally not readily available when running a CCD optimization directly, because optimization algorithms can fail for technical reasons (inadequate parameters, scaling or problem formulation)."*

***Overview of specific aspects***

*My judgments here are based on my current understanding of the work.*

*Does the paper address relevant scientific questions within the scope of WES?*

*Yes. Reducing computational cost of wind energy system design optimization.*

*Does the paper present novel concepts, ideas, tools, or data?*

*Yes. The specifics of the methodology proposed, i.e., the approximations, are new in this context.*

*Is the paper of broad international interest?*

*Yes. All wind energy system designers could benefit from significant advances in this area.*

*Are clear objectives and/or hypotheses put forward?*

*Yes. The reduction of computational cost of wind energy system design.*

*Are the scientific methods valid and clear outlined to be reproduced?*

*Partial. The general overview of the methodology and its application to the case study are clear, but its details are not.*

*Are analyses and assumptions valid?*

*Yes. The analysis is set up well and much care is taken to discuss the assumptions made and their limitations.*

*Are the presented results sufficient to support the interpretations and associated discussion?*

*Partial. The analysis of accuracy of the approximations seems quite solid, but the analysis of computational cost is insufficient.*

*Is the discussion relevant and backed up?*

*Mostly. The discussion is mostly conceptual, but includes at least one statement that would require further explanation ("We can expect … reduced benefits.").*

*Are accurate conclusions reached based on the presented results and discussion?*

*Partial. Statements about computational cost are insufficiently backed by presented results.*

*Do the authors give proper credit to related and relevant work and clearly indicate their own original contribution?*

*Yes. They seem to do a good job of mentioning related relevant literature.*

*Does the title clearly reflect the contents of the paper and is it informative?*

*Partial. It mentions CCD and design optimization, but not anything about the nature of the approximations or the concrete case study. (It is my opinion that a more informative title would be desirable. For example: "A second order approximation for investigating control co-design relevance applied to wind turbine tower design")*

*Does the abstract provide a concise and complete summary, including quantitative results?*

*Yes. Computational cost discussion may need to be amended.*

*Is the overall presentation well structured?*

*Partial. See specific comments.*

*Is the paper written concisely and to the point?*

*Partial. See specific comments.*

*Is the language fluent, precise, and grammatically correct?*

*Mostly. Any remaining issues can be easily fixed by the journal's copy-editors.*

*Are the figures and tables useful and all necessary?*

*Yes. More could/should be added; see specific comments. (N.B.: Figure 1 deserves explicit praise.)*

*Are mathematical formulae, symbols, abbreviations, and units correctly defined and used according to the author guidelines?*

*Mostly. Some may need to be introduced earlier/better; see technical corrections.*

*Should any parts of the paper (text, formulae, figures, tables) be clarified, reduced, combined, or eliminated?*

*Yes. See general comments above and specific comments.*

*Are the number and quality of references appropriate?*

*Yes.*

*Is the amount and quality of supplementary material appropriate and of added value?*

*Yes. The appendices with details about the approximations are actually necessary if no external reference makes their content easily accessible.*

**Authors:** Thank you for this evaluation of the manuscript. We have addressed these aspects in the specific comments and technical corrections below. In addition, the title has been adjusted following this comment and Reviewer 1's comment: *"A sensitivity-based estimation method for investigating control co-design relevance"*

***Specific comments***

- *Paper presentation, focus, and clarity.*

  *As mentioned, the paper's current structure can cause confusion with the reader when trying to understand how the concrete optimization problems (10-12) are reduced to the abstract one (1|2). Namely, in Sec. 3.1, 3.3, 3.4 the case study problem is described. In Sec. 3.2, the reduction to the abstract problem is attempted. I think it can be clearer when Sec. 3 only focuses on presenting the concrete optimization problem and then a new section after that focuses on the reduction of the concrete problem to the abstract version. This new section should be far more elaborate than the current Sec. 3.2 and really explicitly make the correspondence between the f and g of the abstract problem and the constraints and objective of the concrete problem, making sure that it is clear how the two-layered optimization structure of the concrete problem is gotten rid of.*

  *(You could even consider moving the concrete problem first and the abstract problem second, but that is a matter of taste.)*

  **Authors:** Thank you for raising this issue. We agree with this comment. In order to clarify how the concrete problem relates to the abstract one, we have followed the recommendation in this comment and have added a section describing how the estimation formula is applied to the tower problem (Section 4). In addition, the description of the surrogate function for the CoE is described in more detail.

- *Figures and tables*
  *Figure 6 is very illuminating. It would be good to add such a figure as well for the first-order approximation, so that the difference between the two approximations becomes clearer. Furthermore, the full CCD problem will have given rise to a decent number of LCOE-evaluations and would allow to also visualize the actual underlying LCOE-surface for both tower types. It would be of great value if this were done, as it would give a feeling how far or close the approximations (+surrogate) are from the 'ground truth'.*
  **Authors:** We agree with this comment. We have updated Figure 6 to include the first-order estimate function and corresponding estimated optimum. In addition, Table 2 is updated to include the results for the first-order estimation.

Regarding the validity of the CoE estimate functions, the two CCD optimizations evaluate the CoE 12 and 25 times for the standard and soft-soft configurations, respectively. The value of the tower height and control tuning for these iterations is highlighted in Figure S1 below. However, it would be difficult to construct an accurate CoE surface from these point distributions. Instead, we compare in Figure S2 below the value of the CoE calculated during the optimization and the value of the first-order and high-order estimate functions at the same points. Both estimators capture the order of magnitude of the CoE, however the high-order estimator is much more precise. In order to keep the article concise, these figures have not been included in the revised manuscript.

[Figure]

*Figure S1: Path of the optimization algorithm in the design space for the standard and soft-soft configurations.*

[Figure]

*Figure S2: Change of CoE during the optimization process and corresponding value calculated with the estimators.*

**Technical corrections**

- *p1l11: control-co design → control co-design*
  **Authors:** Corrected.
- *p1l20: optimal production → optimal energy production*
  **Authors:** Corrected.
- *All units should be typeset according to the standards, i.e., with a space between number and unit. For example, p2l28: 13MW → 13 MW.*
  **Authors:** Corrected.

- *avoid double parentheses around citations by absorbing by properly using citation macros. For example, p2l29: (e.g. Zahle et al. (2016)) → (e.g. Zahle et al., 2019) [check macros such as \citet and \citep]*
  **Authors:** Corrected.
- *p2l37: "A promising problem for CCD applications is likely to be sensitive to control tuning.": this is a central assumption that requires more justification*
  **Authors:** We have added the following sentence with a reference to a review paper in order to address this comment: (l. 37-38) "Indeed, an integrated design approach is recommended when the physical system and control system are strongly coupled (Allison and Herber, 2014)."
- *p3l70: at each iteration → at each iteration of the optimization algorithm*
  **Authors:** Corrected.
- *p3l76: "If Problem 2 can benefit from a CCD formulation": this makes no sense, as Problem 2 does not depend on c (only on a constant c_r); likely you want to reformulate this*
  **Authors**: We believe that this is explained immediately below, where we explain that the method works by perturbing the value of c_r
- *p3l78-84: The mathematical notation used here needs to be introduced more elaborately and in a more structured way, so in advance of its usage. Moreover, there should be some discussion of the meaning of dx\*, as the natural thing to do would be to consider x(c_r) and x(cr+dc), but the latter is effectively replaced by x(c_r) + dx\* (I am not yet fully convinced that using dx\* isn't introducing some implicit assumptions)*
  **Authors**: we are not sure we completely understand this comment. We believe that the perturbation analysis of the solution is developed in a correct and consistent way, as also demonstrated by the model problem studied in the appendix.
- *p3l84/p4l90: Define 'stationarity condition'/'stationary point of the objective function' formally/explicitly (I guess it is there where the gradient is zero?).*
  **Authors:** The term "stationarity condition" is standard in the field of optimization, and refers to the name of one of the first-order optimality conditions. It is defined explicitly in Eq. 6. In addition, we have added the definition of stationary point on Line 92 (the gradient of the function is zero).
- *p4l97: f(x,c_r,λ): what is the λ doing there?*
  **Authors:** Sentence corrected, the objective function does not depend on lambda here.
- *p4l102-103: Why not mention assumptions explicitly? The current formulation is vague.*
  **Authors:** The assumptions related to the KKT conditions are outside the scope of this paper, and can be found in relevant textbooks. Therefore, the sentence was removed in the revised manuscript.
- *p4l112: It would be easier to follow if the explanation of the figure is in-text and the caption is just the title of the figure.*
  **Authors:** The caption of the figure has been simplified, and the explanation has been moved to the main text.
- *p5l115-116: Why not mention assumptions explicitly? The current formulation is vague. (Likely you mention them after Eq. 9, but then the connection should be made explicit.)*
  **Authors:** The validity assumptions for the first-order estimator is made more explicit by writing it right after Eq. 9.
- *p5l120: finite → small? infinitesimal?*
  **Authors:** Corrected.

- *p5l121-122: Again 'validity assumptions' are not made explicit (it may be as easy as referring back explicitly to some lines above)*
  **Authors:** The validity assumption for the high-order estimator are made more explicit by writing them right after Eq. 10 and as a list.
- *p5Fig1caption: define 'coupling' formally/explicitly in the text before this aspect of the figure is discussed*
  **Authors:** Corrected
- *p6Eq11: δf should be defined before being used*
  **Authors:** Corrected
- *p6l144: noted m → denoted by m*
  **Authors:** Corrected.
- *p6l147: discussion of marginal effect of control on AEP is vague/informal; can you make it more formal/explicit?*
  **Authors:** We have clarified the sentence: (l. 222-223) "However, the net annual energy production is mostly driven by the tower height, whereas the impact of the controller tuning and the inner tower design is marginal in comparison"
- *p7l163-164: "Therefore, the estimator in Eq. (9) is defined using this constraint only and applied to the tower mass minimization problem.": I think this statement should become more prominent/explicit*
  **Authors:** In the revised manuscript, this sentence has been removed and replaced by Section 4.
- *p7l165: 'As a result': say explicitly that Eq. 9's first term therefore becomes zero.*
  **Authors:** Corrected: (l. 209-210) "The objective function for the considered problem is m(t, d, h) and does not depend on the control parameter. Therefore the first term in the estimator equations is zero: ∇f = ∇m = 0 and Δf (dc) = Δm(dc) = 0."
- *p7l169-170: "because the active set is robust, there is little interaction between constraints, and the objective and constraints tend to be nearly linear around the optimum": this statement really needs some justification/references*
  **Authors**: The importance of the validity assumptions was not clear in the submitted manuscript. We have clarified in the methodology section that a violation of the validity assumption affects primarily the precision of the method. The statement regarding the has been reformulated:

  > (l. 204-208) *"Regarding the validity assumptions of the high-order estimator, a preliminary study on the impact of the control tuning on the fatigue damage constraint ensured the robustness of the active set with the chosen range of control tuning variation. In addition, the objective and constraints can be assumed to be linear in $\vec{x}$ provided the change of design is small. However, the validity assumption related to the coupling is more difficult to prove due to the complexity of the problem considered. Therefore, the high-order estimator may be unprecise."*

- *p7Fig2caption: the information in the caption should be integrated in the text; currently this part is really confusion, also due to the fact that the relevant information is spread over text and caption instead of forming a unified discussion.*
  **Authors:** Figure 2 has been removed from the revised manuscript. The CoE estimate function is described more precisely in Section 4, which should address this comment.
- *p7l173: 'surrogate model': explain better what the role of the true LCOE is, e.g., by expanding Eq. 14 with an extra part '= ...' making the connection explicit*

**Authors:** See previous comment.

- *p9l217-218: 1e-… → 10^{-…}*
  **Authors:** Corrected.

- *p10l237-238: "Adding this constraint also reduces the relative importance of fatigue, reducing the potential for CCD, but also showing why the soft-soft tower has lower mass than the standard configuration.": vague, so make more explicit*
  **Authors:** The sentence has been modified as follows: (l. 254-256) *"The Lagrange multipliers associated with fatigue are one order of magnitude smaller, showing a lower relative importance of these constraints and a reduced potential for CCD compared to the soft-soft case"*

- *p12Table1: Make it explicit that the second-order estimator is considered here*
  **Authors:** Corrected.

- *p12l260-265: This paragraph really needs to be expanded to its own subsection at least, as computational cost quantification and discussion is severely underrepresented in the paper. Also, '12 evaluations' are mentioned here, but shouldn't that be 16 as 4 times 4?*
  **Authors:** A new section dedicated to the computational effort has been added. The calculation of the computational effort for the high-order estimator is also detailed.

- *p14l295-296: "We can expect that including this feature in the controller design would translate into reduced benefits.": clarify*
  **Authors:** This sentence has been removed in the revised manuscript.

- *p16EqB3: dc* → dc*
  **Authors:** Corrected.

- *p16EqB4: f(dc) → f(x*+dx*,dc)?*
  **Authors:** The notation Δf (dc) is defined explicitly in the next line.

- *p17l349-350: "we assume that the constraints that do not depend on x contribute marginally to the change of optimum": is this a reasonable assumption (justify)*
  **Authors:** Upon further examination of the explanation of this validity assumption, it was found that it is not necessary for the high-order estimator and the corresponding proof. This is because the following equation (noted B6 in the manuscript) is valid for all active constraints, and not only active constraints that depend on c:

$$\nabla_x g_i(x^*, c_r)^T dx^* = -\Delta g_i(dc) + o(|dc|^2)$$

The relative contribution of the constraints to the change of optimum is reflected in the Lagrange multipliers. In the submitted manuscript, we included an example in Appendix C showing the impact of this validity assumption. However, the presented results were obtained with an error in the calculation of the high-estimator. Figure S3 presents the updated results, showing that the assumption has no impact on the precision of the high-order estimator. As such, we have removed this assumption in the updated manuscript.

[Figure]

*Figure S3: Comparison of the optimal objective value with the first-order estimator and the high-order estimator for problems where the constraint non-dependent on c interacts to a varying degree with the constraint dependent on c. The higher the value of b, the weaker the interaction with the two types of constraints.*

- *p17EqC1: I was wondering why there is no cross term in x and c included.*
  **Authors:** The cross terms between x and c are included in the objective function, through the matrix **P**. Instead, we have not included couplings in the constraints in this example, in order to keep the appendix short. However, the impact of a coupling in the constraint is likely to be similar to the one of a coupling in the objective function.
- *p18-21: Make it explicit that the titles refer to some assumption whose violation is studied.*
  **Authors:** Corrected.

[revised manuscript text omitted]